# Atmospheric Ducts and Their Electromagnetic Propagation Characteristics in the Northwestern South China Sea

**Ning Yang** [1] , **Debin Su** [2,3] **and Tao Wang** [1,*]

[1] School of Electronics and Communication Engineering, Sun Yat-sen University, Shenzhen 518083, China; yangn58@mail2.sysu.edu.cn
[2] College of Electronic Engineering, Chengdu University of Information Technology, Chengdu 610225, China; sudebin@cuit.edu.cn
[3] Key Laboratory of Atmospheric Sounding, China Meteorological Administration, Chengdu 610225, China
[*] Correspondence: wangtao35@mail.sysu.edu.cn

**Abstract:** The propagation of electromagnetic waves beyond the line of sight can be caused by atmospheric ducts, which are significant concerns in the fields of radar and communication. This paper utilizes data from seven automatic weather stations and five radio-sounding stations to statistically analyze the characteristics of the atmospheric ducts in the northwest region of the South China Sea (SCS). After verifying the practicality of numerical analysis data from NCEP CFSv2 and ECMWF in studying atmospheric ducts using measured data, we analyzed the spatial–temporal distribution characteristics of the height of the regional evaporation duct and the bottom height of the elevated duct. The study found that the NCEP CFSv2 data accurately capture the evaporation duct height and duct occurrence rate in the study area, and the elevated duct bottom height calculated from ERA5 and the measured data have good consistency. The occurrence rate and height of the evaporation duct in coastal stations in the northwest of the SCS vary significantly by month, demonstrating clear monthly distribution patterns; conversely, changes in the Xisha station are minimal, indicating good temporal uniformity. For lower atmospheric ducts, the difference in occurrence rates between 00:00 and 12:00 (UTC) is negligible. The occurrence probability of elevated ducts in the Beibu Gulf area is relatively high, mainly concentrated from January to April, and the Xisha area is dominated by surface ducts without foundation layers, mainly concentrated from June to August. Monsoons play a critical role in the generation and evolution of atmospheric ducts in the northwest of the SCS, with the height of the evaporation duct increasing and the bottom height of the elevated duct decreasing after the onset of the summer monsoon. In the end, we simulated electromagnetic propagation loss under different frequencies and radiation elevation angles in various duct environments within a typical atmospheric duct structure.

**Keywords:** atmospheric ducts; northwestern SCS; parabolic equation model; propagation loss

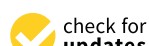



## 1. Introduction

The atmospheric layer has a significant impact on electromagnetic wave propagation, mainly through the refraction and scattering effects of the atmosphere on electromagnetic waves. Variations in temperature, humidity, and air pressure in the atmosphere can lead to changes in the refractive index of the atmosphere, which is a crucial parameter that determines the propagation characteristics of radio waves in the lower atmosphere. Atmospheric ducts are a common occurrence of super-refractive propagation conditions that greatly affect radar and communication equipment at sea [1–3]. Electromagnetic waves that propagate in atmospheric ducts are trapped in the duct layer, resulting in lower path loss for trapped signals and enabling beyond-line-of-sight communication [4–8]. The formation of an atmospheric duct propagation has a significant impact on radar detection. It can cause large differences in radar ranging, angle measurement, and speed measurement,

as well as enhance radar clutter, making radar over-the-horizon detection and reception accompanied by large-area detection blind spots.

Atmospheric ducts can be categorized into an evaporation duct, a surface duct, and an elevated duct, with the surface duct and the elevated duct collectively referred to as lower atmospheric ducts. The evaporation duct is formed due to the evaporation of seawater, which leads to a rapid decrease in the water content of the near-sea surface layer with increasing height, resulting in a change in the refractive index lapse rate. The evaporation duct exists almost always over the ocean, but at different heights, generally occurring at heights below 40 m in the atmosphere near the sea surface, and changes with variations in humidity, temperature, and wind speed. The formation and change in lower atmospheric ducts usually correspond to the temperature inversion process caused by atmospheric motion, such as radiation inversion, sinking inversion, advection inversion, frontal inversion, etc. [9–11], for some special lower atmospheric ducts, they can be formed from the sharp drop in humidity [12].

The evaporation duct height (EDH) is typically low; EDH is the main characteristic parameter describing the refractive index of the atmosphere in the evaporation duct. The EDH parameters can be directly measured using meteorological sounding or meteorological gradiometer techniques [13]. However, due to the unique climate environment and surface type of the ocean, there is a serious lack of meteorological observation stations, making it difficult to achieve accurate regionalization and high-temporal resolution detection. Therefore, the evaporation duct is diagnosed using the model developed based on the sea–air interaction similarity theory, which obtains the marine meteorological parameters and sea surface temperature through measurement or reanalysis [14–16]. Reanalysis data, such as the NCEP CFSv2 (National Centers for Environmental Prediction Climate Forecast System Version 2) data set, are often used for marine meteorological parameters, providing the best estimate of the state of the atmosphere and ocean from 2011 to 2022 [17]. The data have been utilized by researchers to analyze the temporal and spatial distribution characteristics of offshore evaporation ducts [18–20]. Commonly used models for evaporation duct diagnosis include the Paulus–Jescke model [21], MGB model [22], XGB model [23], and NPS model [14]. Many researchers have used the NPS model to study EDH in different sea areas [24–26], and the validity of the model has been verified. Lower atmospheric ducts usually occur at heights below 3 km, and their duct parameters can be obtained from meteorological sounding measurements. For lower atmospheric ducts at sea, the ERA5 reanalysis (The Fifth Generation ECMWF Atmospheric Reanalysis) data are often used for analysis. Yinhe Cheng [10], Yong Zhou [27], I. Sirkoval [28], and Axel von Engeln [29] have studied the characteristics of atmospheric ducts in different regions.

Advancements in approximate numerical algorithms and computing speeds have enabled researchers both domestically and internationally to conduct more comprehensive studies on atmospheric ducts. The main models used for this research include the ray tracing model, duct model, parabolic equation (PE) model, and various hybrid methods that combine these models [30,31]. Among these models, the PE model has become the main method for simulating and analyzing the propagation characteristics of radio waves in atmospheric ducts [31–35].

In the northwest region of the SCS, a variety of mesoscale and sub-mesoscale marine processes occur, including fronts and upwelling, etc. [36,37]. The atmospheric duct environment in this area is complex. As maritime traffic in this region continues to increase, the demand for mobile communication and radar is also growing. However, the propagation of electromagnetic waves is seriously affected by the occurrence of atmospheric ducts, and there is a lack of comprehensive studies on the characteristics of atmospheric ducts in this area. This paper, therefore, uses data from automatic weather stations (AWS) and sounding stations to statistically analyze the characteristics of evaporation ducts and lower atmospheric ducts at the corresponding stations and verifies the reliability of the NCEP CFSv2 and ERA5 numerical analysis data sets (see Section 2.1.2 for an introduction to the data sets) in the calculation of evaporation ducts and lower atmospheric ducts. By

taking the statistical analysis results of atmospheric ducts at the site as prior information to conduct an electromagnetic propagation simulation analysis of ducting events (including hybrid ducting events), the PE model is used with typical ducting parameters to determine the electromagnetic propagation properties in various atmospheric duct environments, including uniform and non-uniform. Figure 1 displays the specific research method used.

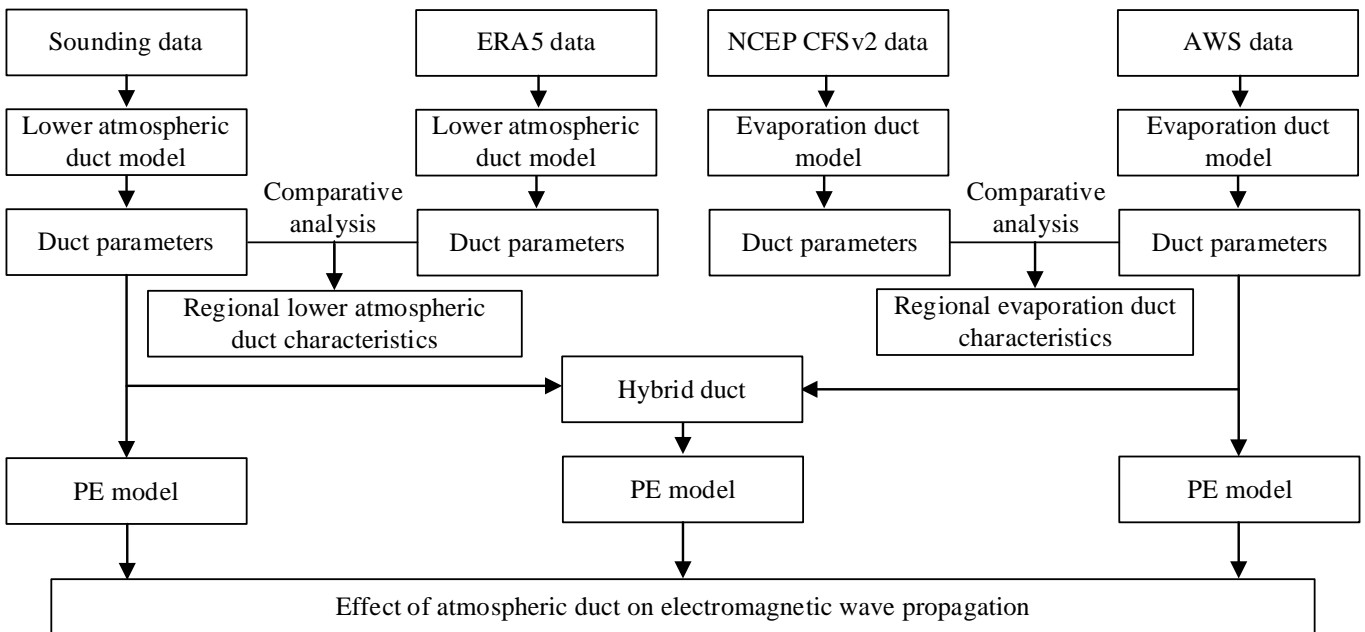

**Figure 1.** The research flow chart of this paper.

## 2. Data and Methods

### 2.1. Data

#### 2.1.1. Observation Data

Meteorological soundings can provide high-resolution data products on the vertical structure of atmospheric temperature, humidity, pressure, and wind, while AWS can provide high temporal-resolution meteorological elements on the ground. In this paper, we use AWS and sounding data from the Wyoming State Weather Network (http://weather.uwyo.edu/ accessed on 12 December 2022) to statistically analyze the characteristics of the evaporation ducts and lower atmospheric ducts of the corresponding stations. Data from 7 AWS (4 coastal stations, 1 Beibu Gulf island, and 2 Xisha islands) and 4 sounding stations (2 coastal stations, 1 Beibu Gulf island, and 1 Xisha island) are used in this study. The specific locations, station numbers (Figure 2), and data information of each observation station are shown in Table 1. For AWS data, station 48,839 has sea surface temperature (SST) observations. Stations 48,845 and 48,839 of the radiosonde stations have almost no observations at 12:00. Unless otherwise specified, all times in this paper are in universal time.

To simplify the correspondence between station numbers and geographical locations, we have adopted the following abbreviations: XS for Xisha, BG for Beibu Gulf, BGE for eastern Beibu Gulf, BGN for northern Beibu Gulf, IPC for central Indochina Peninsula, IPN for northern Indochina Peninsula, and HNC for Hainan (Table 1).

**Table 1.** Data information of AWS and sounding stations.

| Data Source | Station No | Longitude | Latitude | Meteorological Elements | Data Year Span |
|---|---|---|---|---|---|
| AWS | 59,981 (XS) | 112.333°E | 16.833°N | sea level pressure, station pressure, wind speed, temperature, dew point | 2012–2022 |
| | 48,839 (BG) | 107.717°E | 20.133°N | sea level pressure, station pressure, wind speed, temperature, dew point, SST | 2011–2022 |
| | 59,985 (XS) | 111.617°E | 16.533°N | sea level pressure, station pressure, wind speed, temperature, dew point | 2011–2022 |
| | 59,838 (BGE) | 108.617°E | 19.100°N | | |
| | 48,848 (IPC) | 106.600°E | 17.483°N | | |
| | 59,644 (BGN) | 109.100°E | 21.483°N | | |
| | 48,845 (IPN) | 105.671°E | 18.737°N | | |
| Sounding stations | 59,981 (XS) | 112.333°E | 16.833°N | pressure, altitude, temperature, dew point | 2010–2022 |
| | 48,855 (IPS) | 108.200°E | 16.030°N | | 2010–2022 |
| | 48,845 (IPN) | 105.671°E | 18.737°N | | 2014–2022 |
| | 48,839 (BG) | 107.717°E | 20.133°N | | 2013–2022 |
| | 59,758 (HNC) | 110.350°E | 20.030°N | | 2010–2022 |

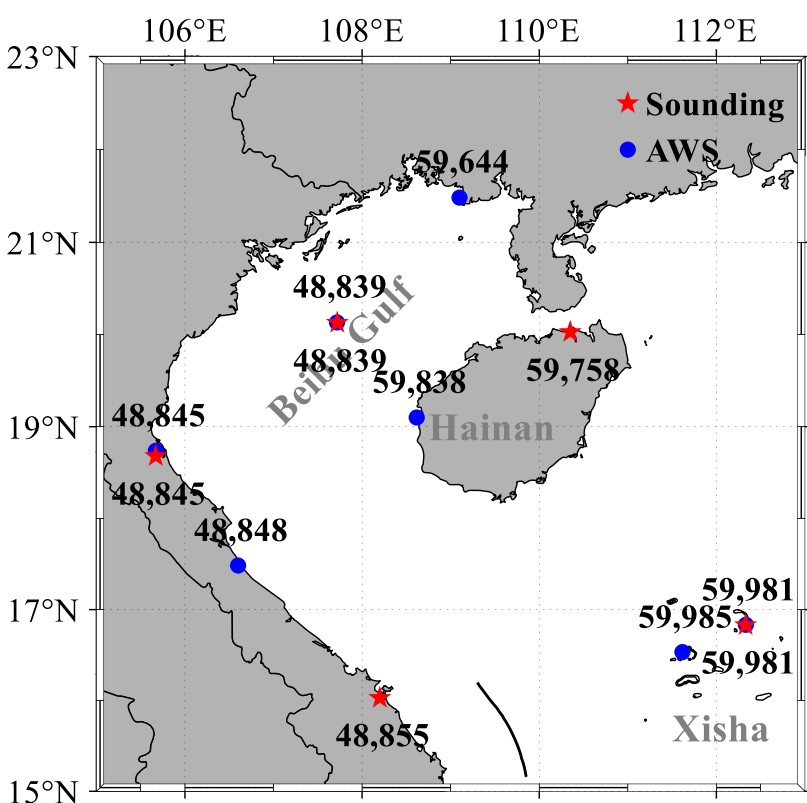

**Figure 2.** The research area of this paper (the red star indicates the position of the sounding station, the blue circle indicates the AWS site, and the corresponding colored number indicates the station number).

### 2.1.2. Numerical Analysis Data

In coastal areas, atmospheric ducts display noticeable horizontal non-uniformity [6,20,37,38]. NCEP CFSv2 and ERA5 data, with their superior spatial resolution, offer a more accurate representation of the distribution features of horizontal non-uniformity in atmospheric ducts along the sea-land boundary region, as compared with other data sources (e.g., MERRA, JRA, and NCEP2, etc.). Several researchers have used this dataset to study atmospheric ducting phenomena in different regions [10,18,20,29].

ERA5 is the fifth-generation reanalysis of global climate and weather by ECMWF (European Center for Medium-Range Weather Forecasts) in the past 8 years. The current ERA5 hourly pressure layer data starts from 1940, with a total of 37 pressure layers, and the data cover the whole world scope. In this paper, the ranges of longitude and latitude

are 105°E∼113°E and 15°N∼23°N, and the data from 2010 to 2022 are used to analyze the lower atmospheric ducts in the northwest of the SCS. The vertical pressure layer of the data is 400∼1000 hPa, and the meteorological elements include temperature, specific humidity, potential, and horizontal wind; see Table 2 for details.

**Table 2.** ERA5 data information used in this article.

| Item | Content |
| --- | --- |
| Data Type | Conventional longitude and latitude grid data |
| Temporal Span | 2010∼2022 |
| Horizontal Resolution | 0.25° × 0.25° |
| Vertical Pressure Layer (hPa) | 400, 450, 500, 550, 600, 650, 700, 750, 775, 800, 825, 850,875, 900, 925, 950, 975, 1000 |
| Time Resolution | 1 h |
| Spatial Range | 105°E∼113°E, 15°N∼23°N |
| Meteorological Elements | Temperature, Specific humidity, Geopotential, V-component of wind, and U-component of wind |

NCEP CFSv2 is an upgraded version of CFSR (Climate Forecast System Reanalysis) [17]. This system assimilates global atmospheric data from land, surface, ships, radiosondes, aircraft, satellites, etc., and can better reproduce the true state of the atmosphere. The data in the range of 105°E∼123°E and 15°N∼23°N for longitude and latitude, respectively, and from 2011 to 2022 are used to analyze the evaporation duct in the northwest of the SCS in this paper. The specific data information is shown in Table 3.

**Table 3.** NCEP CFSv2 data information used in this paper.

| Item | Content |
| --- | --- |
| Data Type | Conventional longitude and latitude grid data |
| Time Span | 2011∼2022 |
| Horizontal Resolution | 0.205° × 0.205° |
| Spatial Range | 105°E∼113°E, 15°N∼23°N |
| Meteorological Elements | Surface pressure, Specific humidity of 2 m, Temperature of 2 m, Surface temperature, U-component of wind at 10 m, V-component of wind at 10 m |

### 2.2. Atmospheric Ducts Model

The atmospheric ducts are stronger super-refractive propagation conditions, which are closely related to atmospheric temperature, humidity, and air pressure. The relationship between refractive index to temperature, air pressure, and water vapor pressure be expressed as follows [39]:

$$N = \frac{77.6}{T}\left(P + 4810\frac{e}{T}\right) \tag{1}$$

where $N$ represents the atmospheric refractive index (N-units), $T$ is the atmospheric thermodynamic temperature (K), $P$ is the atmospheric pressure (hPa), and $e$ is water vapor pressure (hPa). The water vapor partial pressure $e$ can be converted from the specific humidity:

$$e = \frac{qP}{\epsilon + (1 - \epsilon)q} \tag{2}$$

where $q$ is the specific humidity (kg/kg) and $\epsilon$ is a constant (usually 0.622). When considering the curvature of the earth, the modified refractivity can be expressed as follows:

$$M = N + \frac{h}{R_e} \times 10^6 = N + 0.157h \tag{3}$$

where $M$ is usually called modified refractivity and its unit is M-unit, $R_e$ is the average radius of the earth (taken as 6371 km), and $h$ is the altitude above sea level (m).

Tropospheric ducts have different types and spatial structure characteristics, as shown in Figure 3, which presents the corresponding geometric features and parameters of dif-

ferent types of ducts. The structural characteristic quantities of each duct type mainly include the trapping layer top height ($h_t$), trapping layer bottom height ($h_c$), duct bottom height ($h_b$), duct thickness ($h_t - h_b$), trapping layer thickness ($h_t - h_c$), and duct strength ($M_2 - M_1$). The duct strength and trapping layer thickness are important factors that determine whether electromagnetic waves are trapped. In this paper, we record the duct phenomenon at each time as a duct frequency, i.e., the occurrence rate of the atmospheric ducts is the percentage of the duct frequency in the statistical temporal frequency. The duct height ranges from 5 to 40 m as a frequency of duct occurrence for the evaporation duct at a time.

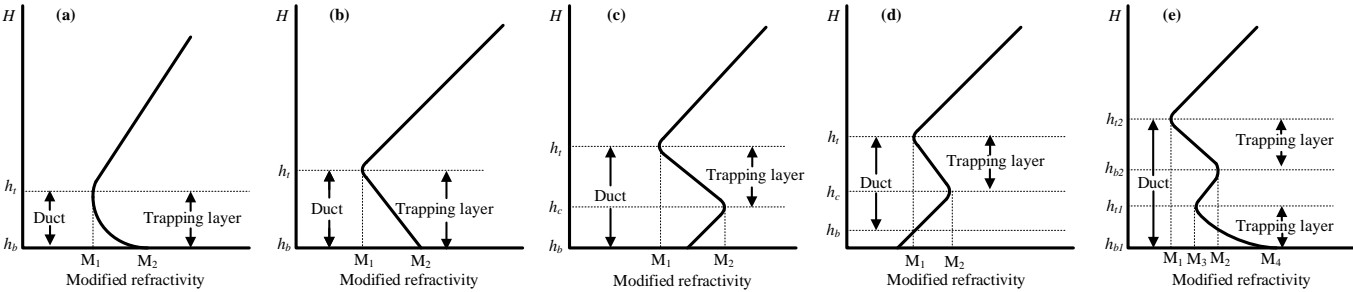

**Figure 3.** Geometric characteristics and parameters of atmospheric ducts: (**a**) evaporation duct; (**b**) surface duct without a base layer; (**c**) surface duct with the base layer; (**d**) elevated duct; (**e**) hybrid duct; $h_t$ is trapping layer top height, $h_c$ is trapping layer bottom height, and $h_b$ is duct bottom height.

### 2.2.1. Evaporation Duct Model

For the evaporation duct, the minimum trapping frequency of electromagnetic waves can be expressed using the vertical decrease rate of EDH and $M$ as follows [40]:

$$f_{min} = 12 \times 10^{10} \cdot \left( -\frac{dM}{dh} \right)^{-0.5} \cdot \eta^{-1.5} \qquad (4)$$

where $dM/dh$ is the vertical gradient of $M$ and $\eta$ is the EDH. The single-parameter evaporation duct model can be represented as follows [21]:

$$M(z) = M_2 + c_0 \left( z - h_t \ln \frac{z + h_0}{h_0} \right) \qquad (5)$$

where $M$ and $M_2$ are called modified refractivity and modified refractivity at height 0 m (380 M-unit is taken in this paper), $c_0$ is the neutral-layered evaporation duct parameter (usually 0.13 M-units/m), $h_t$ is EDH, $h_0$ is the roughness factor (usually 0.00015 m), and the geometric characteristic is shown in Figure 3a.

### 2.2.2. Lower Atmospheric Duct Model

We only simulate the electromagnetic propagation of the lower atmospheric duct structures in Figure 3b,e. So, we present these two lower atmospheric duct models in this paper.

For the surface duct without a base layer (Figure 3b), the duct model can be expressed as follows:

$$M(z) = \begin{cases} M_2 - \dfrac{z(M_2 - M_1)}{h_t - h_b} & z < h_t \\ M_1 + c_1(z - h_t) & z \geq h_t \end{cases} \qquad (6)$$

where $c_1$ is a constant (usually 0.12 M-units/m); $z$ represents the height of the vertical scale; and other parameters correspond to Figure 3b, where $h_b$ is the height of the ground or sea surface ($h_b$ = 0 m).

For the hybrid duct of specific geometric characteristics in Figure 3e, the duct model can be expressed as follows:

$$
M(z) = \begin{cases}
M_4 + c_0\left(z - h_{t1}\ln\dfrac{z + h_0}{h_0}\right) & z \le h_{t1} \\[2ex]
M_3 + \dfrac{z(M_2 - M_3)}{h_{b2} - h_{t1}} & h_{t1} < z \le h_{b2} \\[2ex]
M_2 - \dfrac{z(M_2 - M_1)}{h_{t2} - h_{b2}} & h_{b2} < z \le h_{t2} \\[2ex]
M_1 + c_1(z - h_{t2}) & z > h_{t2}
\end{cases}
\tag{7}
$$

when $z \le h_{t1}$, the meanings of each parameter correspond to Formula (5) ($M_4$ corresponds to $M_2$, and $h_{t1}$ corresponds to $h_t$), $c_1$ is a constant (usually 0.12 M-units/m), and $h_{b1}$ is the height of the ground or sea surface ($h_{b1} = 0$ m).

### 2.3. Evaporation Duct Diagnostic Model

The NPS evaporation duct model uses air temperature, relative humidity, wind speed, pressure, and sea surface temperature at a certain height or at different heights on the sea surface as input parameters and calculates EDH based on the Monin–Obukhov similarity theory. The model first obtains the temperature, humidity, and air pressure and then obtains the atmospheric refractive index profile of the evaporation duct according to the relationship between the atmospheric refractive index, temperature, humidity, and atmospheric pressure. The EDH is then determined by correcting the position of the minimum value of $M$. The vertical section of near-surface temperature $T$ and specific humidity $q$ in the model can be expressed as follows [40]:

$$
T(z) = T_0 + \frac{\theta_*}{\kappa}\left[\ln(\frac{z}{z_0t}) - \psi_h(\frac{z}{L})\right] - \eta_d z
\tag{8}
$$

$$
q(z) = q_0 + \frac{q_*}{\kappa}\left[\ln(\frac{z}{z_0t}) - \psi_h(\frac{z}{L})\right]
\tag{9}
$$

where $T(z)$ and $q(z)$ are the air temperature and specific humidity at height $z$; $T_0$, and $q_0$ are the sea surface temperature and specific humidity when considering the influence of seawater salinity relative humidity; $q_0 = 0.98 q_s(T_0)$ and $q_s(T_0)$ are the sea surface saturation specific humidity calculated based on the sea surface temperature; $\theta_*$ and $q_*$, are, respectively, the characteristic scales of potential temperature $\theta$ and specific humidity $q$; $\kappa$ is the Karman constant; $z_0t$ is the temperature roughness height; $\psi_h$ is the temperature universal function; $\eta_d$ is the dry adiabatic lapse rate, approximately equal to 0.00976 K/m; $L$ represents the Obukhov length; and the water vapor pressure profile can be determined from the functional relationship between water vapor pressure $e$ and specific humidity $q$, that is, Formula (2).

### 2.4. Parabolic Equation Model

The PE is an approximation of the wave equation, which can be used to describe the propagation in a conical area concentrated in a certain direction. In the Cartesian coordinate system, the x, y, z axes represent the wave propagation direction and the horizontal and vertical directions, respectively. When the atmospheric refractive index of the propagation medium is $n$, the two-dimensional scalar wave equation can be expressed as follows:

$$
\frac{\partial^2 \varphi}{\partial x^2} + \frac{\partial^2 \varphi}{\partial z^2} + k^2 n^2 \varphi = 0
\tag{10}
$$

where $k = 2\pi/\lambda$ is the wave number in free space ($\lambda$ is the wavelength of the electromagnetic wave), $\varphi$ represents the electric or magnetic field polarized horizontally or vertically.

If the attenuation function related to x is $u(x, z) = e^{-ikx}\varphi(x, z)$, then the wave equation becomes [41]:

$$\frac{\partial^2 u}{\partial x^2} + 2ik\frac{\partial u}{\partial x} + \frac{\partial^2 u}{\partial z^2} + k^2(n^2 - 1)u = 0 \tag{11}$$

After factorization, the equations for the forward propagation and backward propagation about the coordinate x are obtained, respectively:

$$\frac{\partial u}{\partial x} = -ik(1 - Q)u \tag{12}$$

$$\frac{\partial u}{\partial x} = -ik(1 + Q)u \tag{13}$$

where $Q$ is a pseudo-differential operator:

$$Q = \sqrt{\frac{1}{k^2}\frac{\partial^2}{\partial z^2} + n^2} \tag{14}$$

At this time, the forward propagation solution can be written as:

$$u(x + \Delta x, z) = e^{ik\Delta x(-1+Q)}u(x, z) \tag{15}$$

when the backward propagation is ignored in the standard PE, the operator $Q$ is approximated by using the first-order Taylor expansion, yielding the standard PE as follows [41]:

$$\frac{\partial^2 u(x, z)}{\partial x^2} + 2ik\frac{\partial u(x, z)}{\partial x} + k^2(n^2 - 1)u(x, z) = 0 \tag{16}$$

Due to the complexity of the atmospheric environment, Formula (16) is difficult to solve analytically. In this case, a numerical solution to the electromagnetic field problem can be obtained via numerical methods. The commonly used numerical solutions of the PE model include the Split Step Fourier Transform (SSFT) algorithm and the finite difference (FD) method. The SSFT solutions [41] of the narrow-angle and wide-angle parabolic equations are

$$u(x + \Delta x, z) = \exp\left[ik(n^2 - 1)\frac{\Delta x}{2}\right]F^{-1}\left\{\exp\left[-ip^2\frac{\Delta x}{2k}\right]F\{u(x, z)\}\right\} \tag{17}$$

$$u(x + \Delta x, z) = \exp[ik(n - 1)\Delta x] \times F^{-1}\left\{\exp\left[-ip^2\frac{\Delta x}{k}\left(\sqrt{1 - \frac{p^2}{k^2}} + 1\right)^{-1}\right] \times F\{u(x, z)\}\right\} \tag{18}$$

$F$ and $F^{-1}$ represent the Fourier transform and the inverse Fourier transform, respectively, and $p = k\sin\theta$ ($\theta$ is the electromagnetic wave propagation angle). In radio-wave applications, two parameters are of interest: propagation factor and propagation loss (usually called path loss). The path loss is the ratio between the power radiated by the transmitter antenna and the power available at a point in space, which can be determined as follows:

$$PL = -20\log|u| + 20\log(4\pi) + 10\log x - 30\log\lambda \tag{19}$$

## 3. Results and Discussion

### 3.1. Evaporation Duct

Compared with the meteorological observations available at other stations, AWS 48,839 (BG) adds a unique SST element (Table 1). Therefore, we compare and analyze the SST data of this station with the data from NCEP CFSv2 to evaluate its accuracy. Figure 4 shows the daily average of SST observation differences at 00:00, 06:00, 12:00, and 18:00 at AWS 48,839 (BG) and the NCEP CFSv2 SST from 2011 to 2022 (SST_dif = AWS-NCEP).

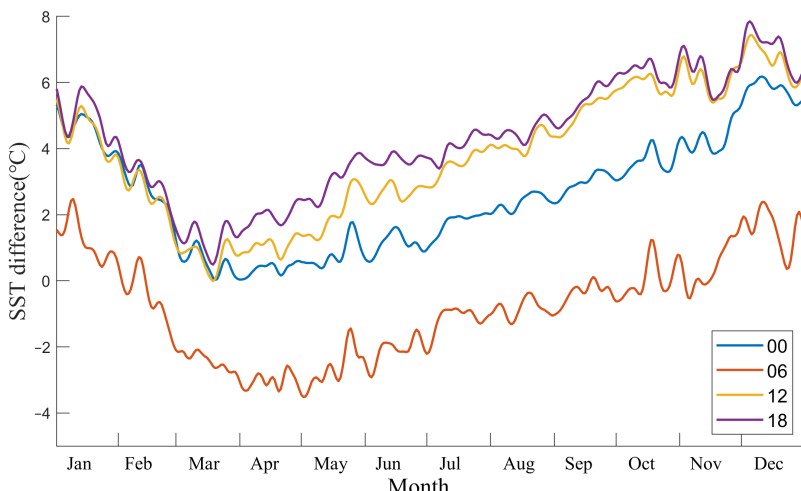

**Figure 4.** The daily average of observation difference between 48,839 (BG) and NCEP CFSv2 SST at 00:00, 06:00, 12:00, and 18:00 from 2011 to 2022.

From Figure 4, it can be observed that the SST difference shows a trend of first decreasing and then increasing. The AWS SST values are consistently higher than those of the numerical analysis at 00:00, 12:00, and 18:00. The smallest difference value is observed in April at 06:00, and the AWS SST results in most months are smaller than those of the numerical analysis. Overall, the total average differences of SST at 00:00, 06:00, 12:00, and 18:00 are 2.6 °C, −0.8 °C, 3.8 °C, and 4.3 °C, respectively. The SST difference at 06:00 is mostly within the range of −3~2 °C, while the differences at the other three time points are in the range of 0~7 °C. So, the SST is underestimated at 00:00, 12:00, and 18:00, which increases the actual air–sea temperature difference and reduces the occurrence rate and height of evaporation ducts.

Based on the analysis, it can be concluded that the NCEP CFSv2 SST at 06:00 is relatively close to the actual observation. In addition, the daily variation in evaporation duct intensity and height in the northern part of the SCS generally follows a low–high–low pattern, with both intensity and height reaching their maximum at 06:00 in the afternoon [42]. During this period, electromagnetic waves are more susceptible to interference. Therefore, for other stations that lack SST observations, the NCEP CFSv2 SST data at 06:00 can be used as a substitute. We only analyze the characteristics of EDH at each site at 06:00, and the incidence of the evaporation duct is calculated using the observation data at 00:00, 06:00, 12:00, and 18:00.

### 3.1.1. Characteristics of Evaporating Ducts

As the evaporation ducts in coastal area differs significantly from those in inland seas [3,18,43], we divided the seven AWS into two categories: four coastal stations (48,855 (IPS), 48,848 (IPC), 59,644 (BGN), and 59,838 (BGE)) and three island observation stations (48,839 (BG), 59,981 (XS), and 59,985 (XS)). Figure 5 illustrates the occurrence rate of evaporation ducts in coastal stations (left) and the monthly average distribution of EDH at 06:00 (right). From the monthly average distribution of evaporation duct occurrence rates, it can be observed that the occurrence probability characteristics of evaporation ducts in coastal stations 48,855 (IPS), 48,848 (IPC), and 59,838 (BGE) are relatively consistent. The occurrence rates of evaporation ducts at these three sites continued to decrease from January to March of the following year and then began to increase after April, with a decrease in November during the increase process. In contrast, at 59,644 (BGN), the incidence rate continued to decrease from December to August of the following year and then continued to increase after August. Overall, the maximum occurrence probability of evaporation ducts in coastal stations was in December, and all stations exhibited significant monthly

non-uniformity characteristics, indicating that the occurrence probability of evaporation ducts varies significantly across different months.

The monthly average distribution of EDH at 06:00 indicates that the characteristics of 48,855 (IPS), 48,848 (IPC), and 59,838 (BGE) EDH are relatively consistent. The average EDH from January to December exhibits a phenomenon of first increasing and then decreasing, while the 59,644 (BGN) shows a double fluctuation of increasing, decreasing, increasing, and decreasing from April to October. From the monthly average variation in EDH, the annual EDH of 48,845 (IPN) is relatively stable, with small monthly differences. The EDH of this station is relatively low throughout the year, with an average EDH of less than 10 m. The EDH of 59,838 (BGE) has significant monthly differences, and the average EDH of the station throughout the year is greater than 10 m.

From the above analysis, it can be concluded that 48,845 (IPN) has a relatively low probability of duct occurrence and duct height, while 59,838 (BGE) has a relatively high probability of duct occurrence and duct height, followed by 59,644 (BGN). Therefore, the probability of evaporation duct occurrence and the duct height exhibit obvious spatial and temporal distribution characteristics. This is mainly because the formation mechanism of the evaporation duct is closely related to the turbulence in the marine atmospheric boundary layer, especially at the sea–land junction. The formation mechanism and changes in evaporation ducts are more complex in such areas.

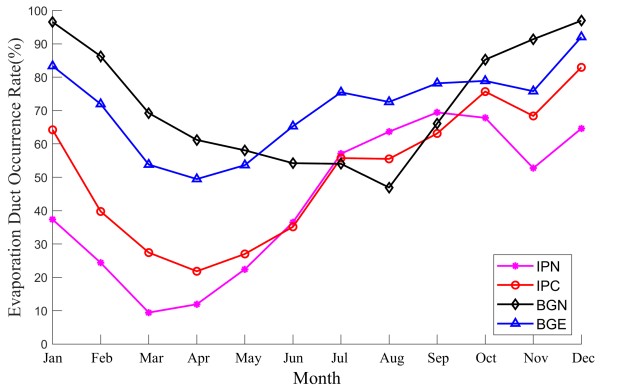 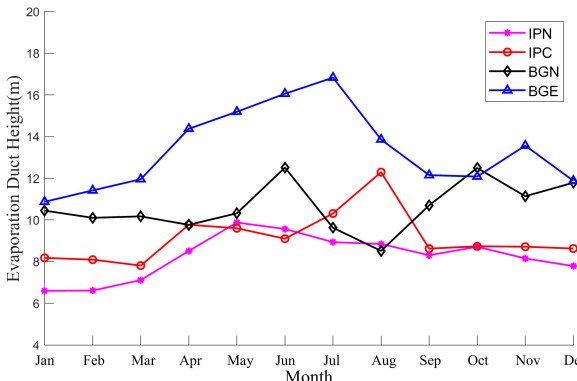

**Figure 5.** The evaporation duct occurrence rate (**left**) and the monthly average distribution of EDH at 06:00 (**right**) in coastal AWS.

Figure 6 displays the occurrence rate of evaporation ducts (left) and the monthly average distribution of EDH at 06:00 (right) for inland sea islands. The occurrence rate of evaporation ducts in 48,839 (BG) is similar to that of coastal stations, decreasing from December to March/April of the following year and increasing after April. This is due to the station being surrounded by continents on three sides, which is significantly affected by land airflow. In the Xisha area, the monthly duct incidence rate of 59,981 (XS) and 59,985 (XS) remained stable, with annual average duct incidence rates of 73.9% and 71.6%, respectively, and the monthly average incidence rate remained above 60%. However, the duct incidence rate in 48,839 (BG) was relatively low, with an annual average duct incidence rate of 28.7%. Therefore, the occurrence rate of evaporation ducts in the two regions exhibited notable spatial differences.

Based on the monthly average distribution of duct height at 06:00, the changing trend of 48,839 (BG) is relatively similar to that of coastal stations, characterized by an increase from January to a peak and then a decrease towards December. The annual average EDH for 48,839 (BG) is 9.47 m, and the monthly average distribution characteristics of EDH are consistent with the research by Yang Shi et al. [19]. In contrast, the EDH in the Xisha area fluctuates frequently but within a smaller range, and the annual duct height is relatively high (with annual average EDHs of 15.4 m and 14.0 m for 59,981 (XS) and 59,985 (XS), respectively), which are typically maintained at a height of over 10 m. Despite being far

away from land and relatively close to each other, significant differences in the occurrence rate and height of duct between the 59,981 (XS) and 59,985 (XS) were observed in June and July.

From the analysis, it can be concluded that the monthly average difference in the occurrence probability of evaporation ducts is smaller in areas less affected by land airflow, and the height of the evaporation duct is higher. Furthermore, the occurrence probability and height of evaporation ducts in different non-coastal areas exhibit notable differences, primarily due to the main mechanism of duct formation being the physical process of seawater evaporation, which is significantly affected by wind speed, sea surface temperature, and other factors, resulting in temporal and spatial inhomogeneity of the evaporation duct.

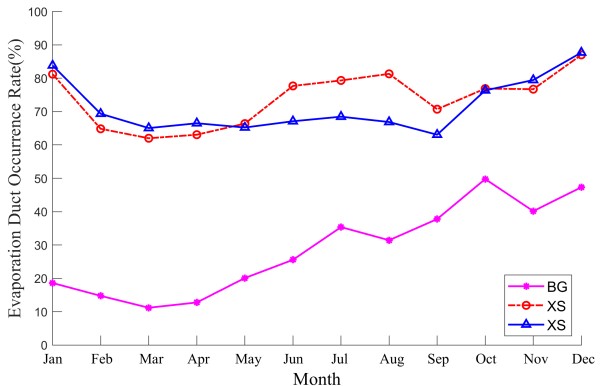 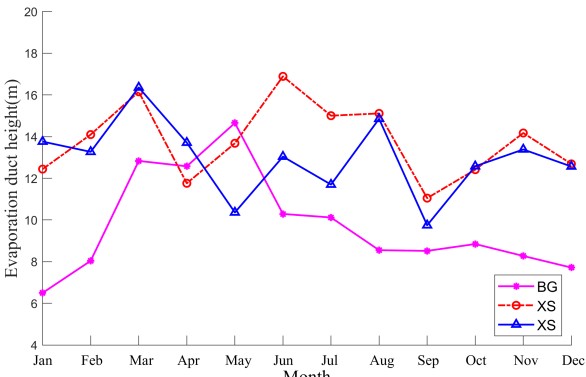

**Figure 6.** The evaporation duct occurrence rate (**left**) and monthly average distribution of EDH at 06:00 (**right**) in inland sea AWS.

To further demonstrate the applicability of NCEP CFSv2 data in analyzing evaporation ducts, we compared the incidence of the duct at 48,839 (BG) with the monthly mean distribution of EDH at 06:00 (as shown in Figure 7). The results show that the difference in duct occurrence probability is relatively large (up to 26%) from January to April, while the difference from July to December is small. In terms of monthly average duct height, the difference in EDH is relatively large (up to 5 m) from March to May, while there is less variation in duct height in other months. Overall, the annual average duct probability difference is 6%, and the average relative difference of EDH is 2 m at 06:00. This indicates that the evaporation duct characteristics obtained from NCEP CFSv2 data can reflect the actual evaporation duct phenomenon to a certain extent and have a good reference value.

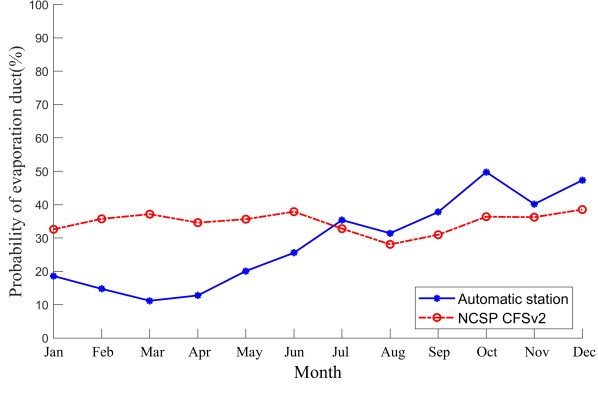 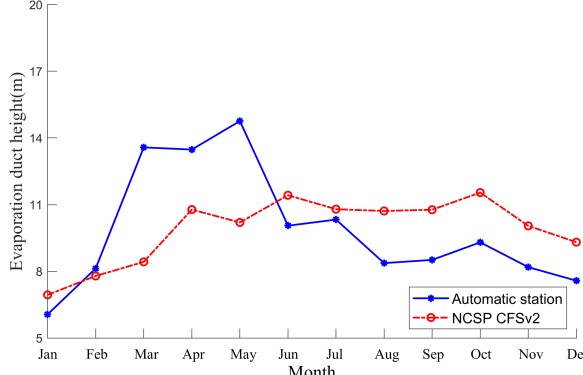

**Figure 7.** The evaporation duct occurrence rate (**left**) and the monthly average distribution of EDH of 48,839 (BG) at 06:00 (**right**).

Equation (4) shows that EDH is an important factor affecting electromagnetic wave trapping. Therefore, we used the NCEP CFSv2 data from 2011 to 2022 to calculate the monthly average distribution of EDH at 06:00 in the northwestern SCS (Figure 8). In terms of time distribution, the height of the duct gradually decreased from October to April of the following year. March/April had the smallest EDH in the whole year. However, after April, the overall EDH in the northwest of the SCS suddenly increased, and the monthly average of EDH at 06:00 was larger from May to October, with the overall area maintaining a height above 15 m. The main reason for this phenomenon was the outbreak of the SCS summer monsoon, which led to an increase in temperature and enhanced evaporation. This, in turn, increased EDH.

In October, the SCS winter monsoon broke out, the temperature decreased, and seawater evaporation weakened, leading to a decrease in EDH. Regarding spatial distribution, the coastal area of Hainan, China had the highest EDH in the northwest of the SCS throughout the year. Four months before the outbreak of the summer monsoon, a high-value area, the "strip", gradually appeared in the area between Hainan, China, and the Indochina Peninsula, which was mainly due to the prevailing northwest wind, which brought warm wind from the southeast alongside high-temperature and high-humidity water vapor, forming a "long strip" duct high-value area.

After the summer monsoon arrived in the Beibu Gulf region, the EDH increased in May/June and the EDH high-value area in this region began to decrease after June. The EDH in the coastal areas of the Indochina Peninsula gradually increased after February and lasted until September. There was a high–low–high variation characteristic of EDH when extending from the coastal area to the inland sea, and Hainan, China, also had this feature, except for the northern coastal area.

In summary, this study reveals that the occurrence of evaporative ducts and their EDH exhibit significant variations in different coastal regions and over different months. The Xisha region shows a relatively stable monthly probability of evaporative duct occurrence, but a high annual average value of EDH. Monsoons have a profound impact on the occurrence of evaporative ducts in the northwest region of the SCS, with a noticeable increase in EDH during the summer monsoon and a decrease during the winter monsoon. These findings provide valuable insights into the spatial and temporal distribution of evaporative ducts in coastal areas, which can aid in the development of effective strategies for marine navigation and communication.

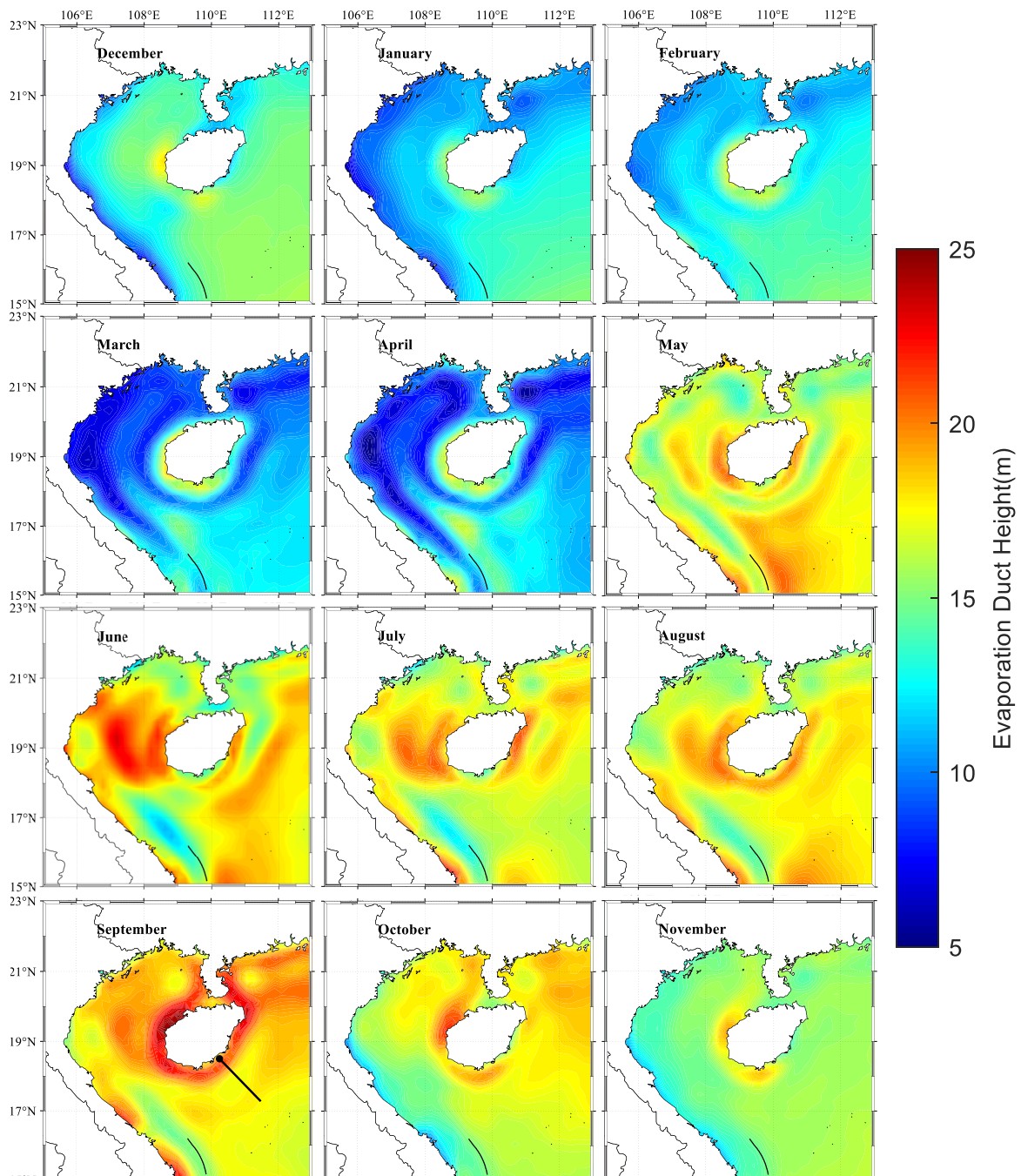

**Figure 8.** The monthly average distribution of EDH at 06:00 in the northwest of the SCS (where the black dots in September indicate the position of the radiation source, and the lines indicate the path of electromagnetic wave radiation; please refer to Section 3.1.2).

### 3.1.2. Electromagnetic Propagation Characteristics in Evaporation Duct Environment

In atmospheric duct propagation, electromagnetic waves are trapped in the atmosphere, resulting in lower path loss and enabling trans-horizon propagation. However, specific conditions must be met for duct propagation to occur:

(1) An atmospheric stratification junctions with $dM/dz < 0$ must exist at a certain height near the ground layer or boundary layer.

(2) The wavelength of the electromagnetic waves must be smaller than the maximum trapping wavelength $\lambda_{max}$ (the frequency must be higher than the lowest trapping frequency $f_{min}$).

(3) Electromagnetic radiation sources situated within the atmospheric duct layer have a higher probability of being trapped. In the case of an elevated duct, the radiation source may sometimes be located below the bottom of the duct, but in this case, the radiation source must be close to the duct bottom and the duct must be very strong.

(4) The radiation elevation angle of electromagnetic waves must be less than a certain critical elevation angle.

Therefore, we considered transmission frequencies of 1 GHz, 3 GHz, 6 GHz, and 9 GHz, and used a parabolic antenna Gaussian beam for the antenna and beam type. Other radiation source parameters are shown in Table 4. Unless otherwise specified, the radiation source and receiver heights in this paper are both 8 m.

**Table 4.** Radiation source parameters.

| Radiation Source Parameters | Specific Settings |
| --- | --- |
| Frequency | 1 GHz, 3 GHz, 6 GHz, 9 GHz |
| Antenna and receiving height | 8 m, 8 m |
| Antenna elevation angle | 0°, 0.5°, 1°, 2°, 3° |
| Polarization mode | Horizontal polarization |
| Horizontal and vertical range | 0~200 km, 0~300 m |
| Antenna and beam type | Parabolic antenna Gaussian beam |
| 3dB beamwidth | 2° |

From Section 3.1.1, it can be seen that the annual average EDH around Xisha is about 15 m, so we set the EDH to 15 m; the corresponding evaporation-duct-modified refractivity vertical section and the propagation loss when the electromagnetic wave frequency is 9 GHz and the elevation angle is 0.5° are shown in Figure 9.

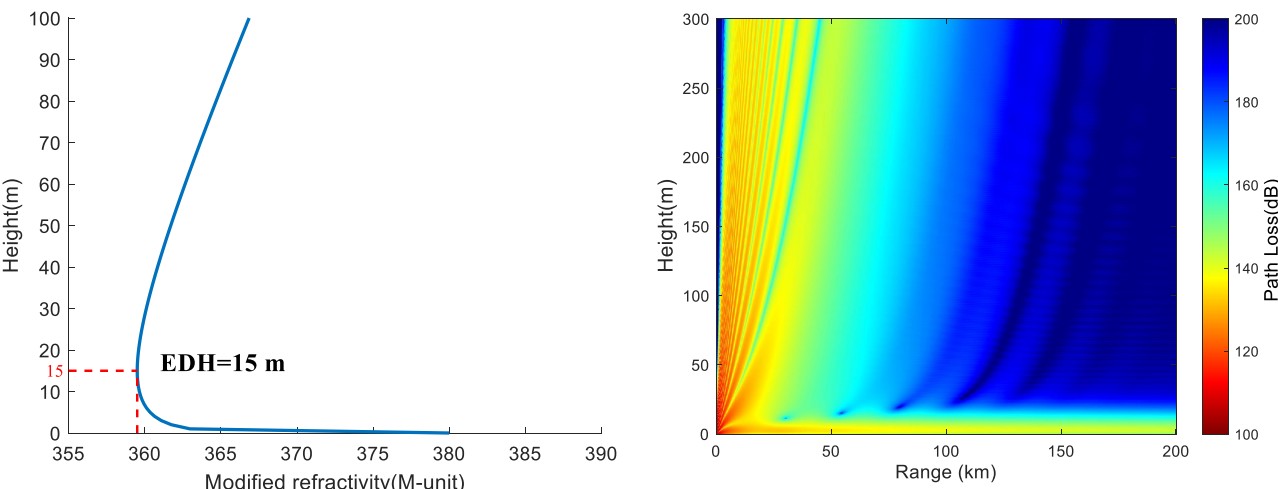

**Figure 9.** The vertical profile of the evaporation duct with modified refractivity (**left**) and the propagation loss when the frequency of the electromagnetic wave radiation source is 9 GHz, and the elevation angle is 0.5° (**right**).

When the EDH is 15 m, the corresponding *M* value is 359.5 M-units. From Figure 9 (right), it is evident that the propagation loss in the EDH layer is lower than that above the height layer, forming a "long strip" structure with a low propagation loss close to the 0 m layer. This indicates that the electromagnetic wave is effectively trapped in the duct layer. As the horizontal distance increases, the propagation loss inside the trapped layer gradually increases, but the reduction is relatively small compared with the propagation loss outside the trapped layer.

Figure 10 displays the propagation loss at different radiation frequencies (left, elevation angle is 0.5°) and elevation angles (right, frequency is 9 GHz). From the left figure, it is

evident that as the frequency increases, the propagation loss decreases. At 6 GHz to 9 GHz, the rate in propagation loss decreases as the distance increases. From the right figure, it can be seen that in the case of a transmission frequency of 9 GHz, the propagation loss continues to increase as the elevation angle of the radiation source increases. However, as the horizontal distance increases, the rate in propagation loss under different elevation angles remains relatively stable and does not show a significant increase. In summary, the higher the frequency of the radiation source and the lower the elevation angle, the easier it is to be trapped by the evaporation duct (special *M* profile structure, such as Figure 9 (left)), thus forming duct propagation.

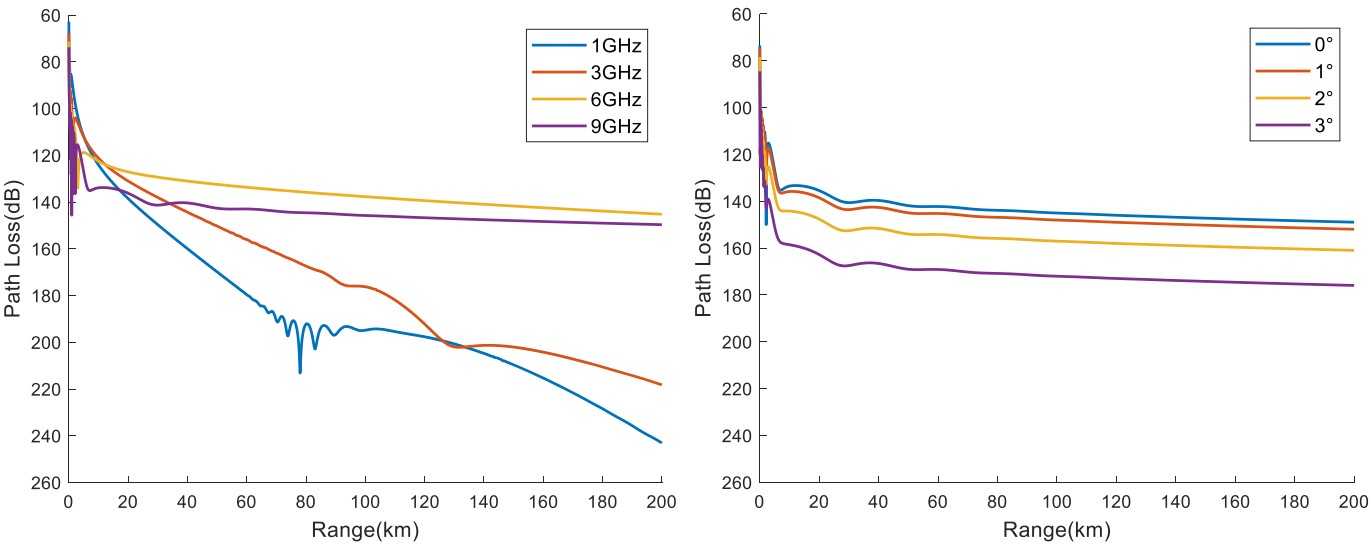

**Figure 10.** Propagation loss at different radiation frequencies (**left**, elevation angle is 0.5°) and elevation angles (**right**, frequency is 9 GHz).

We set the duct height parameters based on the average EDH (black line in Figure 8, longitude and latitude: from 110.25°E, 18.50°N to 111.48°E, 17.27°N) at 06:00 in September, using a certain location in Hainan Province as the radiation source. The other duct parameters are set as shown in Figure 9 (left), and the specific duct height settings are shown in Figure 11a. The red dots represent the distance between each grid and the radiation source, along with its duct height. From Figure 11b, it is evident that electromagnetic waves are effectively trapped within the duct layer, but the difference between uniform and non-uniform evaporation duct environments is not noticeable. Therefore, we simulated the propagation losses at horizontal distances of 15 m and 17 m for EDH (Figure 11c). Within the horizontal range, the propagation losses at EDH = 15 m fluctuate less with distance, while the propagation losses at EDH = 17 m fluctuate more (Figure 11d). When the frequency of the radiation source is 6 GHz and 12 GHz, the fluctuation phenomenon of propagation loss is not obvious (Figure 12), and there is a high propagation loss area at a certain height close to the ground when the frequency is 12 GHz (Figure 12d). Combining Figure 11b, Figure 12b, and Figure 12d, it can be seen that the fluctuation phenomenon of propagation loss is related to the frequency of the radiation source and the height at which it is located.

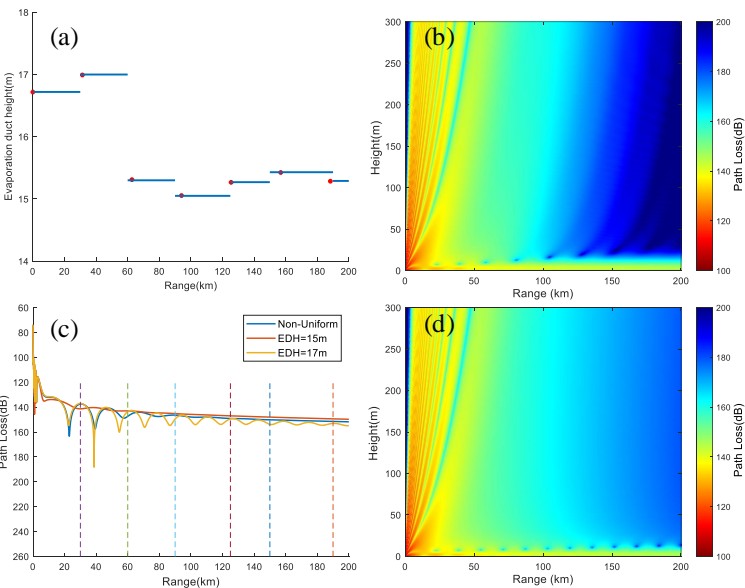

**Figure 11.** Electromagnetic propagation losses in uniform and non-uniform evaporation duct environments. (**a**) Non-uniform EDH setting, the red dots indicate the distance between each grid and the radiation source, and the blue line represents the EDH parameters used within a specific distance range; (**b**) electromagnetic propagation loss in non-uniform duct environment; (**c**) the propagation loss in uniform and non-uniform duct environments with an elevation of 0.5°, a frequency of 9 GHz, and a radiation source. The dashed line represents the horizontal distance boundary between different EDH; (**d**) propagation loss at an elevation of 0.5°, a frequency of 9 GHz, and a duct height of 17 m.

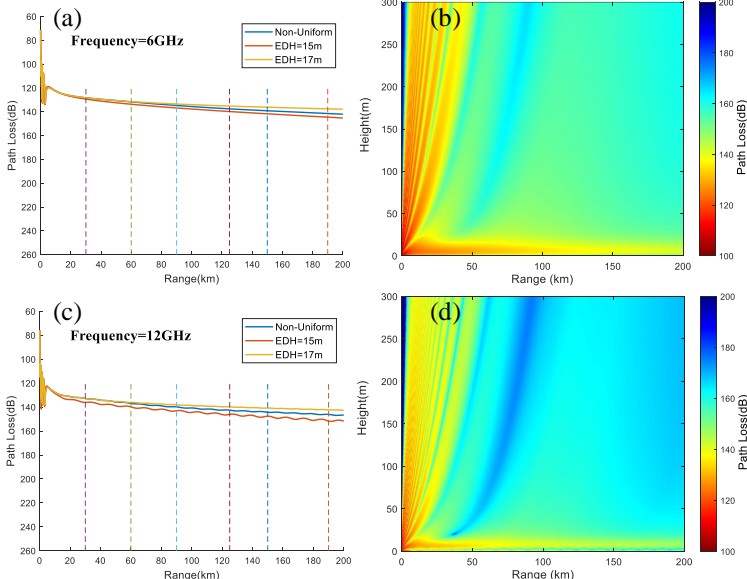

**Figure 12.** The electromagnetic propagation loss in different atmospheric duct environments with radiation sources at frequencies of 6 GHz and 12 GHz: (**a**) electromagnetic propagation loss in uniform and non-uniform atmospheric duct environments at a radiation source frequency of 6 GHz; (**b**) electromagnetic propagation loss in a non-uniform atmospheric duct environment at a radiation source frequency of 6 GHz; (**c**) electromagnetic propagation loss in uniform and non-uniform atmospheric duct environments at a radiation source frequency of 12 GHz; (**d**) electromagnetic propagation loss in a non-uniform atmospheric duct environment at a radiation source frequency of 12 GHz.

*3.2. Lower Atmospheric Ducts*

3.2.1. Characteristics of Lower Atmospheric Ducts

Lower atmospheric ducts can be classified into surface ducts and elevated ducts, with surface ducts being the main cause of over-the-horizon detection for navigation radars. In contrast, evaporation ducts typically only result in weak over-the-horizon detection [4]. Elevated ducts have a relatively small impact on actual electromagnetic wave propagation, mainly due to their inability to meet the conditions for over-the-horizon propagation.

Figure 13 illustrates the monthly average distribution of lower atmospheric duct occurrence rates at the five sounding stations. It is evident from Figure 13 that the lower atmospheric duct occurrence rates at 48,839 (BG), 48,845 (IPN), and 48,855 (IPS) are relatively high. Among them, 48,839 (BG) and 48,845 (IPN) primarily record duct events at 00:00. The monthly average occurrence rate of 59,758 (HNC) is relatively low (less than 20%), mainly due to its proximity to the mainland in both the north and south, which makes it difficult to form a good sea–land breeze circulation. The average occurrence rate of 59,981 (XS) exhibits a clear monthly distribution feature, with the probability first increasing and then decreasing from January to December, and reaching its maximum value in June.

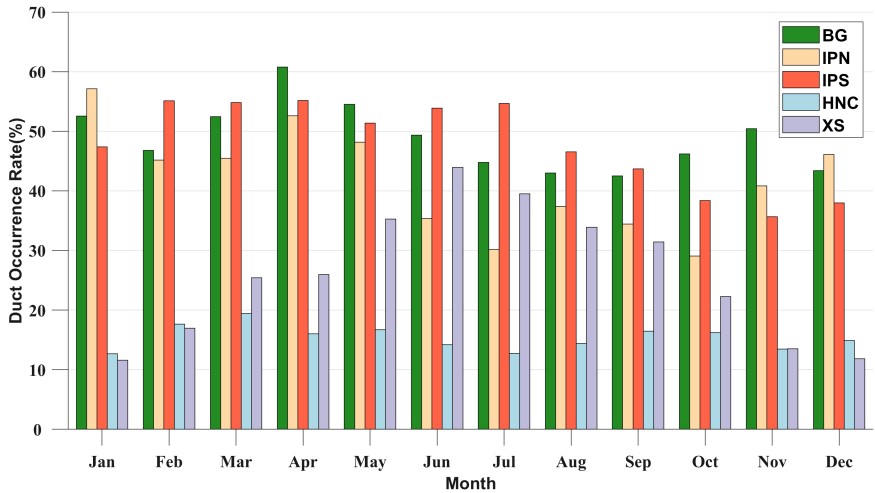

**Figure 13.** Monthly average distribution of lower atmospheric duct incidence in sounding stations.

To distinguish between the surface duct and elevated duct, we analyzed the monthly average incidence of different types of lower atmospheric ducts at each station (Figure 14). From Figure 14 (left), it is evident that surface ducts primarily occur from May to September, which is the time following the onset of the SCS summer monsoon. The probability gradually decreases after the onset of the winter monsoon. Stations 48,855 (IPS) and 59,981 (XS) have the highest probability of surface duct occurrence among the five stations, and the monthly average distribution of occurrence probabilities with monthly variations is relatively similar for the two stations. The monthly average distribution of occurrence probabilities for the other three stations is also relatively consistent.

From Figure 14 (right), it is evident that the three stations with the highest occurrence rates of elevated duct are 48,839 (BG), 48,855 (IPS), and 48,845 (IPN), which exhibit distinct monthly distribution characteristics. That is, the probability of duct occurrence varies significantly with month. The other two stations have lower occurrence rates of elevated duct (less than 20%), but the probability of occurrence of elevated ducts increases slightly with month and tends to stabilize.

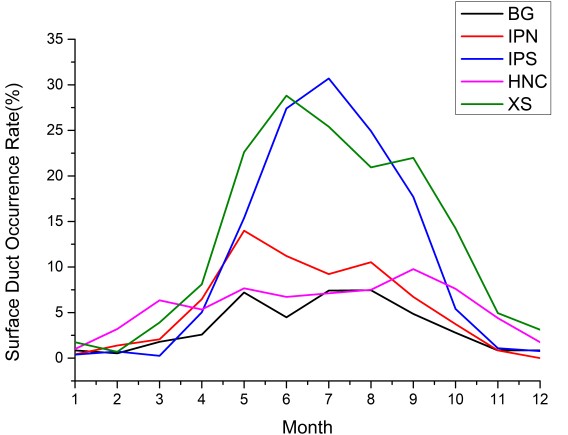 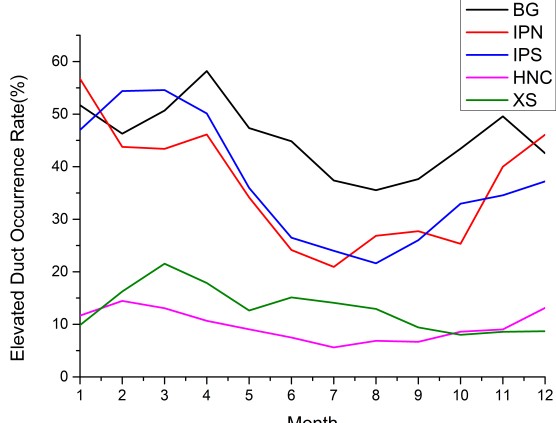

**Figure 14.** The occurrence rate of the surface duct (**left**) and elevated duct (**right**) of the sounding station.

Stations 48,855 (IPS), 59,758 (HNC), and 59,981 (XS) have sufficient sounding observation data at 00:00 and 12:00, allowing us to compare the monthly average occurrence probabilities of the surface duct and elevated duct at these times (Figure 15). In terms of surface duct occurrence rate, the probability of duct occurrence at station 48,855 (IPS) at 00:00 is higher than that at 12:00. The probability of duct occurrence at station 59,981 (XS) is higher before July at 00:00 but lower than that at 12:00 after July. The probability of duct occurrence at station 59,758 (HNC) is higher than that at 12:00 from May to August (i.e., after the onset of the SCS summer monsoon). In terms of the occurrence rate of elevated ducts, the occurrence rate of ducts at stations 48,855 (IPS) and 59,981 (XS) at 12:00 is higher than that at 00:00. However, the difference in the monthly average occurrence rate of ducts at stations 59,758 (HNC) and 59,981 (XS) at 00:00 and 12:00 is not significant. Overall, there is little difference in the occurrence rate of lower atmospheric ducts between 00:00 and 12:00, with a higher probability of occurrence for surface ducts at 00:00 and a higher probability of occurrence for elevated ducts at 12:00.

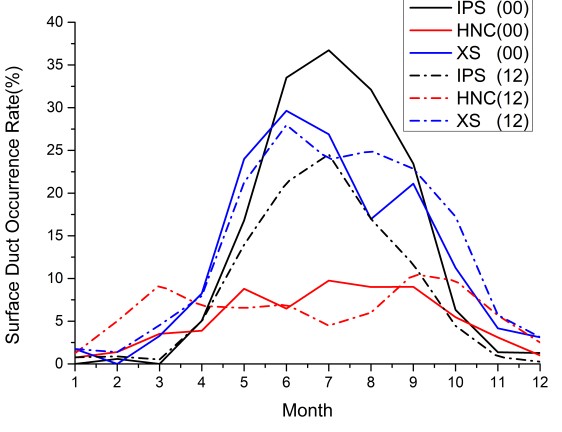 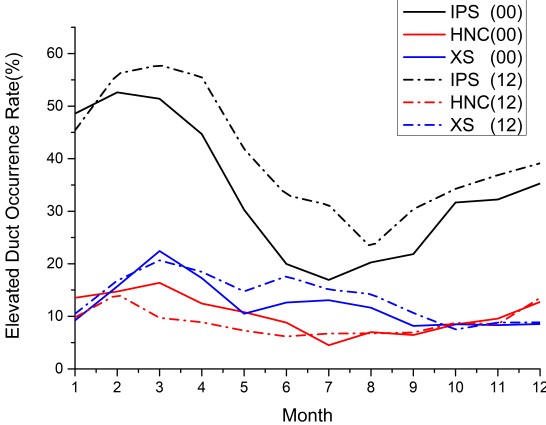

**Figure 15.** Incidence rates of surface duct (**left**) and elevated duct (**right**) at 00:00 and 12:00.

Table 5 displays the percentage of surface ducts without a base layer in the total surface duct. The results indicate that the 59,981 (XS) site has the highest proportion of surface ducts without a base layer, reaching 98.7%, followed by the 48,855 (IPS) site. Overall, the occurrence rate of surface ducts without a base layer is higher than that of surface ducts

with a base layer, indicating that the surface ducts observed at the statistical stations are primarily based on surface ducts without a base layer.

**Table 5.** Percentage of the surface duct without a base layer in the total surface duct.

| Station No | 48,845 (IPN) | 48,839 (BG) | 48,855 (IPS) | 59,758 (HNC) | 59,981 (XS) |
|---|---|---|---|---|---|
| Percentage (Total number of surface duct) | 58.0% (157) | 62.5% (104) | 73.6% (966) | 63.7% (537) | 98.7% (1228) |

Figure 16 presents the statistical distribution of surface duct strength (left) and thickness (right) at each station. The surface ducts are grouped into 0∼30 M-units intervals with a 2 M-units interval and 60∼200 m intervals with a 20 m interval. Ducts larger than 30 M-units and 200 m are grouped separately. From the surface duct strength distribution, it can be observed that surface duct strengths of 48,839 (BG) and 48,855 (IPS) are primarily concentrated between 2 and 8 M-units, with similar probability distributions. The surface duct strengths of 59,981 (XS) and 59,758 (HNC) are mainly concentrated in the range of 2∼4 M-units, and the distribution characteristics of both are more consistent with the months. The surface duct strength of station 48,845 (IPN) is mainly concentrated between 8 and 10 M-units. Regarding surface duct thickness, the primary thickness at each station is below 80 m.

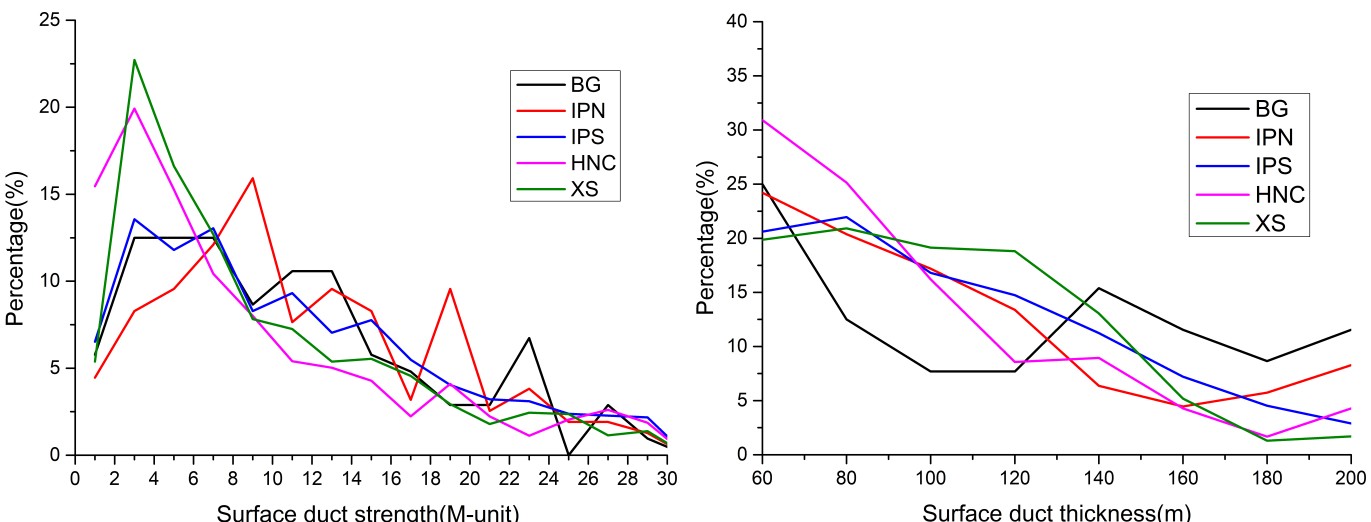

**Figure 16.** Statistical distribution of surface duct strength (**left**) and thickness (**right**).

The bottom height of the elevated duct obtained from the ERA5 data is considered to be relatively close to actual observations [10,29]. In this study, we screened the temperature, humidity, pressure, and other ERA5 data corresponding to the time of elevated duct events in the data of five sounding stations to calculate the elevated duct. Subsequently, we obtained the bottom height of the elevated duct from different data sources. The comparison results between the two sources are shown in Figure 17, indicating that ERA5 data can better reflect the characteristics of the bottom height of the elevated duct; these research results are consistent with the studies of Cheng Y et al. [10].

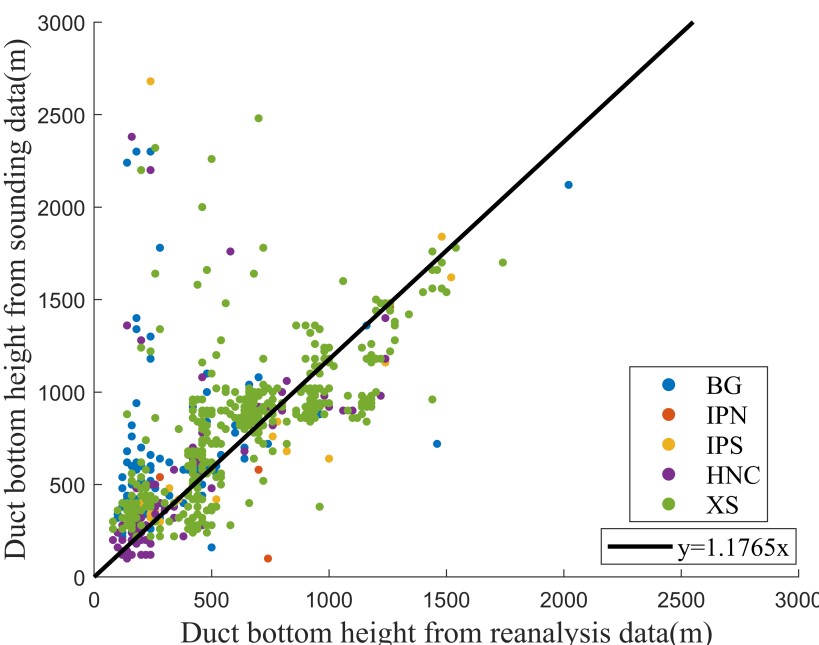

**Figure 17.** Scatter diagram of the bottom height of elevated duct retrieved from sounding data and ERA5 data.

Although ERA5 data may miss certain duct phenomena, it is known to provide a good description of the bottom height of elevated ducts. Figure 18 illustrates the monthly average distribution of the bottom height of the elevated duct in the northwest of the SCS. The results demonstrate that the bottom height of the duct during the onset of the summer monsoon (May to September) is the lowest month of the year, indicating that the arrival of the summer monsoon reduces the bottom height of the elevated duct. Conversely, as the winter winds arrive, the bottom height of the duct begins to increase after October, reaching its maximum value in December, and then gradually decreases until April of the following year. This suggests that the presence of winter winds lifts the height of the elevated duct. These research results are consistent with the studies of Cheng Y et al. [10].

From a spatial distribution perspective, the northeast of the Beibu Gulf displays the highest height of the elevated duct from December to April of the following year. During this time period, the height of the southeast region is greater than that of the northwest region. Additionally, the area connecting the Beibu Gulf and the SCS gradually appears as a low duct with a high bottom from January to April. This phenomenon is mainly due to the prevailing southeast wind, which prevents the elevated duct from lifting well. In summary, the elevated duct exhibits obvious distribution characteristics as time and space change.

To summarize, lower atmospheric ducts are frequently observed in the eastern coast of the Indochina Peninsula and the Beibu Gulf region. In the Xisha region, the occurrence rate of lower atmospheric ducts shows distinct monthly distribution patterns, with monsoons playing a crucial role in the formation and development of suspended ducts in the northwest of the SCS.

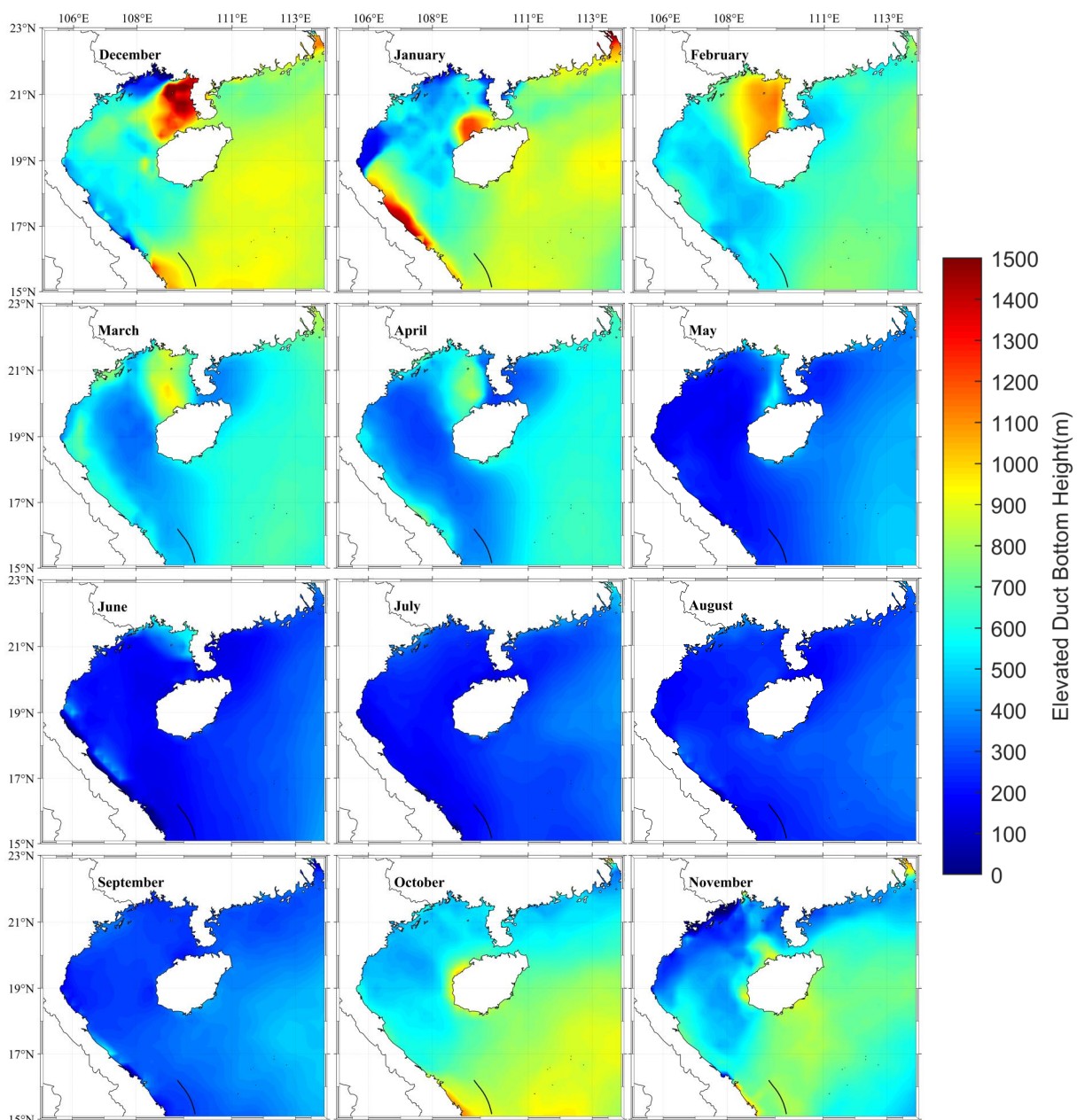

**Figure 18.** Monthly average distribution of the bottom height of the elevated duct in the northwest of the SCS.

### 3.2.2. Electromagnetic Propagation in Surface Duct Environment

As discussed in Section 3.2.1, it is evident that the surface duct without a base layer dominates at station 59,981 (XS), with a high probability of a duct strength of 4 M-units and a duct thickness of 80 m. As a result, we set $M_2$ = 380 M-units, $M_1$ = 376 M-units, $h_t$ = 80 m, and $h_b$ = 0 m (Formula (6)), and the duct structure is depicted in Figure 19 (left) when a height above 80 m is increased by 120 M-units/km. Furthermore, we only considered the propagation loss in the case of radiation source elevation angles of 0°, 0.5°, 1°, 2°, and 4°. The other parameter settings are displayed in Table 4, with a radiation source frequency of 9 GHz and an elevation angle of 0.5°. The variation in electromagnetic propagation loss with distance and height at a 0.5° elevation angle is demonstrated in Figure 19 (right), which clearly indicates that electromagnetic waves are trapped within the duct layer.

Figure 20 displays the electromagnetic propagation loss at a height of 8 m at different frequencies (elevation angle is 0.5°, left) and antenna elevations (9 GHz, right) at a frequency

of 9 GHz. The results indicate that the electromagnetic propagation loss does not decrease with an increase in the horizontal distance at different radiation frequencies. However, the frequency of electromagnetic wave loss jitter increases with an increase in frequency. Moreover, at different elevation angles, it is observed that as the elevation angle increases, the propagation loss increases. The magnitude of the propagation loss reduction also increases when increasing at 1° intervals until it cannot be trapped. In summary, the frequency variation in the surface duct environment has little effect on the electromagnetic propagation loss in the horizontal direction. However, the elevation angle of the radiation source is a critical factor in determining whether the electromagnetic wave is trapped.

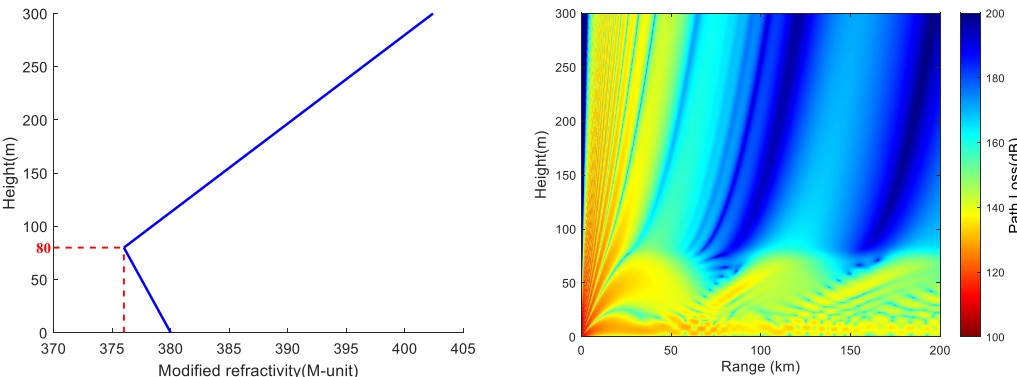

**Figure 19.** The surface duct structure used in the simulation (**left**) and the electromagnetic propagation loss at a radiation source frequency of 9 GHz and an elevation angle of 0.5° (**right**).

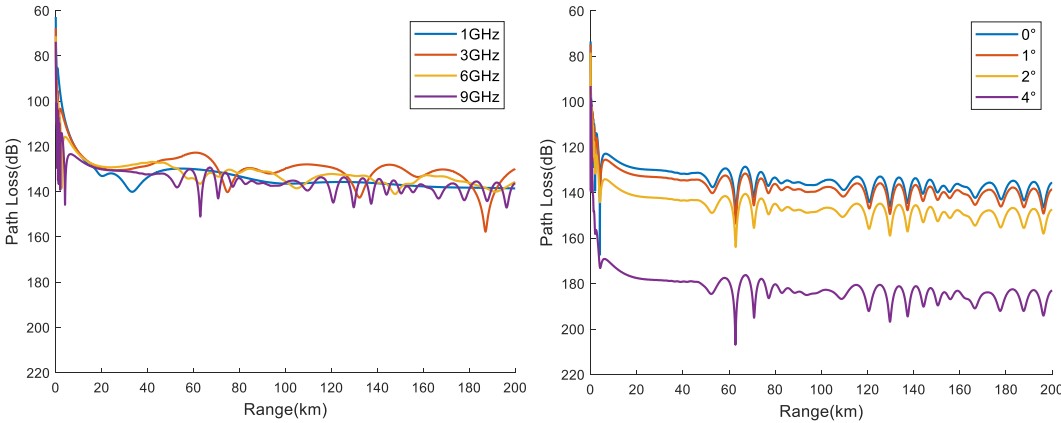

**Figure 20.** Propagation loss at different frequencies (elevation angle is 0.5°, **left**) and antenna elevation angle (frequency is 9 GHz, **right**).

### 3.3. Electromagnetic Propagation in Hybrid Duct Environment

Evaporation ducts are commonly observed in the lower atmospheric structure of the marine atmospheric boundary layer. For instance, in Section 3.1.1, it was found that the average annual incidence of evaporation ducts in the Xisha area is over 70%. Hence, the northwest of the SCS is also susceptible to mixed atmospheric environments of evaporation ducts and surface ducts. Figure 21a illustrates the characteristics of the hybrid duct structure employed in the simulation, with an EDH of 15 m and a surface duct height of 80 m (in Formula (7): $h_{t1} = 15$ m, $h_{b2} = 40$ m, $h_{t2} = 80$ m, $M_1 = 355$ M-units, $M_2 = 360.84$ M-units, and $M_3 = 359.5$ M-units), corresponding to the electromagnetic propagation loss at 9 GHz, and an elevation of 0.5° is shown in Figure 21b. Compared with Figure 19 (left) and Figure 21b, the "sinusoidal wave" frequency of propagation loss in the hybrid duct environment is more frequent, and the propagation loss at a height of 15 m increases faster with distance, indicating a greater electromagnetic propagation loss.

To investigate the electromagnetic propagation losses of radiation sources at an altitude of 8 m and signal reception at different altitudes, we set the signal reception heights to 4 m (below the radiation source height), 10 m (above the radiation source height), 15 m (EDH), and 20 m (above EDH). Figure 21c shows that as the signal reception height increases, the propagation loss also increases, and the fluctuation at 30∼60 km also increases. For radar, areas with high propagation loss can form detection blind spots. Furthermore, when the radiation sources are at heights of 8 m, 15 m, and 20 m, there is no noticeable difference in the vertical scale of propagation loss (Figure 21d). Above a height of 15 m, the propagation loss increases significantly. This phenomenon is characterized by a decrease and subsequent increase in propagation loss from heights of 0∼80 m, with the minimum occurring at a height of 15 m.

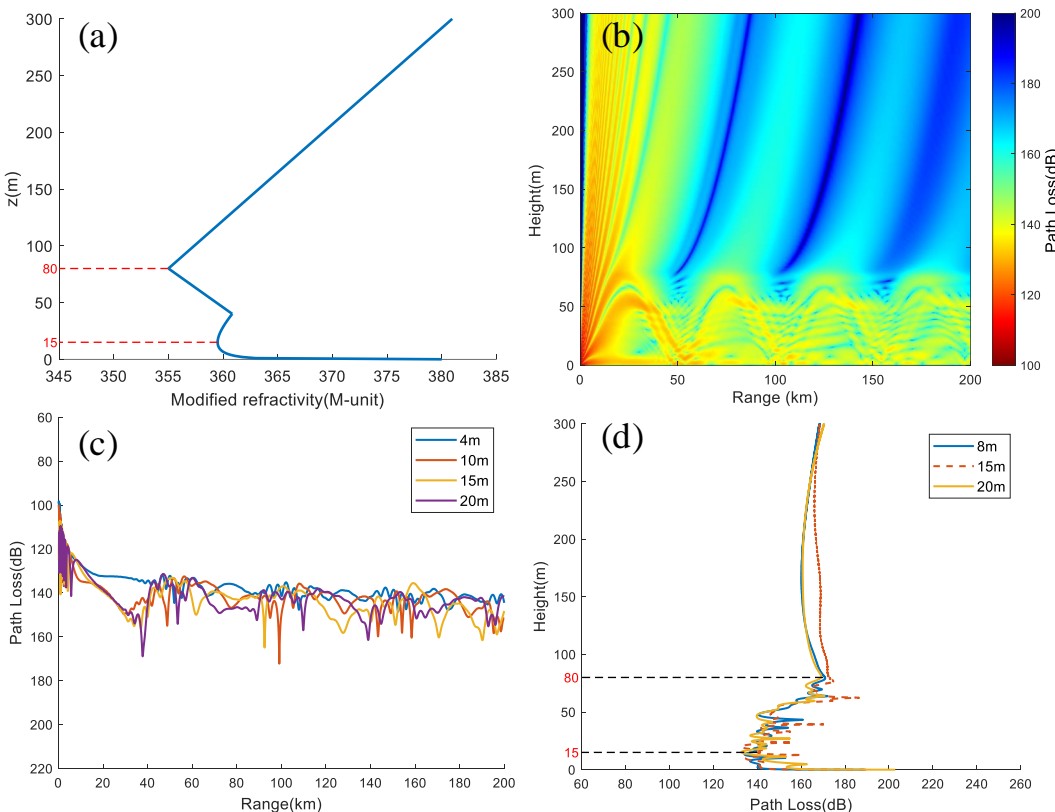

**Figure 21.** Electromagnetic propagation loss in a hybrid duct environment. (**a**) Hybrid duct structure; (**b**) propagation loss at 9 GHz and 0.5° elevation angle; (**c**) propagation loss at different heights (the frequency is 9 GHz, and elevation angle is 0.5°); (**d**) radiation source at different heights and horizontal distances is the vertical propagation loss at 100 km.

## 4. Conclusions

The presence of atmospheric duct environments has altered the path and range of electromagnetic wave propagation, resulting in unique propagation characteristics in communication, detection, navigation, and other systems. The primary conclusions presented in this article are derived from data analysis and electromagnetic propagation simulation. They can be summarized as follows:

(1) The maximum occurrence rate of evaporation ducts at the coastal stations is in December, all have obvious monthly non-uniform distribution characteristics, and EDH also exhibits this feature. The probability and height of evaporation ducts occurring in different coastal areas also have significant differences, with consistent monthly variation trends. Due to the small impact of terrestrial airflow, the monthly average difference in the probability of evaporation duct occurrence at the Xisha Station is small, and the annual

average value of EDH is relatively large. Furthermore, the NCEP CFSv2 data at 06:00 can be utilized for the analysis of evaporation ducts in the SCS region.

(2) The monthly average occurrence rate of lower ducts in the Xisha region shows an increasing trend from January to June and then decreases from July to December. Surface ducts are mainly non-base layer surface ducts. The occurrence rate of lower ducts at other stations does not show significant monthly changes. In terms of time, the probability of surface duct occurrence is relatively high at 00:00, while the incidence of elevated ducts is mainly at 12:00. Furthermore, the ERA5 data can effectively reflect the characteristics of duct bottom height and can be used to analyze the characteristics of low-altitude ducts in the SCS region.

(3) In a uniform evaporation duct environment, a "long strip" structure with low propagation loss is easily formed within the duct layer. The higher the frequency of the radiation source and the lower the elevation angle, the easier it is to be trapped by the evaporation duct. In a uniform surface duct environment, frequency variation has a relatively small impact on electromagnetic propagation loss in the horizontal direction. However, the elevation angle of the radiation source is an important factor in determining whether electromagnetic waves are trapped in this environment. In the hybrid duct environment combining the evaporation duct and surface duct, the "sine wave" frequency of propagation loss is more similar to that in the uniform surface duct. The propagation loss in the EDH within the hybrid duct increases faster with distance.

**Author Contributions:** Conceptualization, N.Y. and T.W.; methodology, N.Y. and T.W.; validation, N.Y. and T.W.; formal analysis, N.Y. and T.W.; investigation, N.Y.; resources, N.Y.; data curation, T.W. and N.Y.; writing—original draft preparation, N.Y.; writing—review and editing, D.S. All authors have read and agreed to the published version of the manuscript.

**Funding:** This research received no external funding.

**Data Availability Statement:** Sounding data and AWS data: http://weather.uwyo.edu/, accessed on 12 December 2022; ERA5 data: https://cds.climate.copernicus.eu/, accessed on 1 November 2022; NCEP CFSv2 data: https://rda.ucar.edu, accessed on 10 November 2022.

**Acknowledgments:** We acknowledge the use of data from the ECMWF team (https://cds.climate.copernicus.eu/, accessed on 1 November 2022), and University of Wyoming (http://weather.uwyo.edu/, accessed on 12 December 2022), and NCAR (https://rda.ucar.edu/, accessed on 10 November 2022); we thank all the editors and reviewers for their valuable comments, which greatly improved the presentation of this paper.

**Conflicts of Interest:** The authors declare no conflict of interest.

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
