# Peer review of "Atmospheric Ducts and Their Electromagnetic Propagation Characteristics in the Northwestern South China Sea"

_remotesensing, doi:10.3390/rs15133317_

Round 1

Reviewer 1 Report

The authors utilized data from seven automatic stations and five radio-sounding stations to statistically analyze the characteristics of the atmospheric ducts in the northwest region of the South China Sea. I have a number of comments and concerns listed below, and the authors should make revisions of this manuscript before it can be potential published.

1.     Why use NCEP CFSv2 and ERA5 data? NCEP CFSv2 is a type of forecasting data, which is different from ERA5. Please add the reasons for using these two datasets relative to others (e.g. MERRA, JRA and NCEP2, etc.)

2.     Why choose the studying period as 2011-2022? To my understanding, a climatic period of 30 years is better. Also, why choose South China Sea as the study region. Please add more descriptions.

3.     It is not necessary for showing 4 decimal places, e.g. in L200, L275, L352.

4.     Please redraw Fig. 5, Fig. 6 and Fig.17. In Fig. 5, the authors should better add the station names? It is difficult for linking the station number to its location. Some other figures also have this problem. In Fig. 6, the colors of two lines are red and too similar. In Fig. 17, the color bars and text are too small. They are the same, and the author may use one large color bar.

5.      Please remove the calculating equations from Result Section to the Method Section.

Author Response

请参阅附件。

Reviewer 2 Report

Please see the above attachment.

Reviewer 3 Report

In this paper, the authors mainly carry out two aspects of work, the first is the statistical analyses of the atmospheric duct data in the northwest of the South China Sea to get the distribution characteristics of the atmospheric duct; the second is the modeling of the propagation loss distribution in different atmospheric environments according to the parabolic equation method. The research results of the first aspect of this paper can be used for reference in the industry and have certain significance. However, the second aspect of the study is not significant, so it may be considered to compress the length.

I believe that the main problems or aspects for improvement are as follows:

(1)  The abstract has too much content and can be reduced to highlight the work of the authors. In addition, the description of the experimental results in the abstract, such as: less impact, higher and other words can be given to the evaluation value, so that the experimental results are clearer.

(2)  In line 54, the physical meaning of evaporation duct height (EDH) can be described, e.g., EDH is the main characteristic parameter describing the refractive index of the atmosphere in evaporation duct.

(3)  The content in lines 76 to 82 contains the negative effects of atmospheric duct on radar systems, which can be placed in the first paragraph of the introduction.

(4)  At the end of the introduction, a section distribution needs to be added.

(5)  The introduction of duct profiles in Chapter 2 (lines 104 to 116) can be put together with the introduction of duct models in Section 2.2, or simply deleted.

(6)  Figure 2 presents the research flowchart of this paper, and I feel that the related contents can be a separate section to present the main research contents of this paper.

(7)  Section 2.2.1 contains the atmospheric refractive index and the modified refractive index, which I feel is inappropriately titled as a lower atmospheric ducts model.

(8)  Equation 17 and related contents can be put into 3.1.2.(2).

(9)  The content in lines 456-458 gives the modeling parameters of the surface duct, and the modeling equation based on the related parameters should be given, as in Equation (18). The modeling expression is also given for the modeling of the hybrid duct refractive index profile in Section 3.3.

(10) Some references are too old. Please consider to update new ISI articles to imporve this section.

A native English speaker is required to polish the paper。

Reviewer 4 Report

The manuscript studies the atmospheric ducts occurrence and the statistical characteristics of different duct types in northwestern part of the South China Sea (SCS). In this aim the authors use local meteorological measurements of seven automatic weather stations and five radio-sounding stations situated on the cost of Beibu gulf and on the Hainan island and adjacent islands. The statistics covers the period between 2011 and 2022. The duct parameters are studied through well known and widely used log-linear profile for evaporation duct, formula (18), and bi- and tri-linear profiles for surface and elevated ducts, Fig. 1. The local measurements are compared to the data coming from NCEP Climate Forecast System version2 (for evaporation ducts) and ECMWF (European Center for Medium-Range Weather Forecasts) Reanalysis v5 (ERA5) for surface and elevated ducts. The manuscript: 1) has shown that the NCEP CFSv2 and ERA5 data is compatible (more or less) with the local measurements and can be used to analyze the characteristics of evaporation duct and lower atmospheric ducts in the SCS region; 2) has reported duct characteristics for the specific area of Beibu gulf.

 Overall impression: the manuscript is unnecessarily long, contains uninformative parts. The sub-sections 3.1.2 and 3.2.2, related to electromagnetic propagation, report no new results; this makes sub-section 2.3 of the parabolic equation redundant as well. Instead, the authors could make comparison with results reported in, for instance, "Spatio-Temporal Distribution of Evaporation Duct for the South China Sea", 2014, Doi: 10.1109/OCEANS-TAIPEI.2014.6964520; “Duct climatology over the South China Sea based on European Center for Medium Range Weather Forecast reanalysis data”, 2021, Doi: 10.1016/j.jastp.2021.105720; or the quoted in the manuscript Reference [17]. This would help to assess the conclusions in the manuscript.

            In addition, there are some repetitions in the text: lines 117-120 repeat the Introduction (lines 104-114 would be better to move to Introduction), lines 137-143 are also a repetition.

Other Recommendations for Authors

1) Fig. 1 (c): in order not to confuse "trapping layer" and "duct thickness", right "trapping layer" instead of just "trapping''

2) lines 47-48 read: "The evaporation duct exists almost always over the ocean, but at different heights, generally occurring at a height of 40 m ..."; in fact, 40 meters are considered the max evaporation duct height, the world average evaporation duct height is much less than 40 m...

3) lines 113-114 read: "the occurrence rate of the atmospheric ducts is the percentage of the duct frequency in the statistical time frequency", this is unclear.

4) Reference [34] does not refer to Bean and Datton; [34] repeats [41].

5) Section 2.2: Rather than repeating known methods, it is better to cite one or two adequate references and say that the manuscript follows them.

6) Provide Reference for formula (17).

7) lines 317-318 read: "The electromagnetic wave radiation source must be located in the atmospheric ducts layer." This is not mandatory.

8) Table 4 what is the beamwidth?

9) Lines 327-330 read: "corrected refractive index" The quantities M and Mo are called Modified refractivity and Modified refractivity at height 0.

10) lines 430-432: "to invert the elevated duct" What do the authors mean under "invert"?

11) Line 457: Fig 18 instead of Figure 17; line 460: Fig.4 instead of Fig. 5

12) Line 470: "... it is observed that as the elevation angle increases, the propagation loss decreases"; according to Fig. 19 (right) is the exact opposite...

13) Sub-section 3.3 reads " Electromagnetic propagation in a homogeneously mixed duct environment"; in fact, this is a medium with more than one M inversion, not a "homogeneously mixed duct".

Reviewer 5 Report

See the attached document

See the attached document

Round 2

Reviewer 1 Report

The authors have revised their manuscript according to my comments and suggestions, and response to the comments correctly too. I think the manuscript can be accepted in current form.

Reviewer 4 Report

The authors have put considerable effort into improving the article but there are still things to fix.

1. In their reply to reviewers, instead of writing "This error has been corrected," authors should insert the exact text of the correction. Also, they should provide the entire revised manuscript with corrections explicitly inserted, not "hidden"... this would help in evaluating the final text.

2. I still believe that Sections related to electromagnetic propagation report no new results and should be removed from the paper; nevertheless, if those Sections and results are kept in order to “serve as a point of reference for conducting electromagnetic propagation loss tests in the South China Sea region in the future” as the authors explained, they should right the correct form of the PE in Eq. 13 and indicate the reference they followed to prepare Section 2.4 Parabolic equation model (not just for SSFT but for the derivation of the PE).

3. In relation to point 2: Table 4. Have you added the 3dB beamwidth?

4. It seems that after page 14 the corrections have not been made: at least I do not see “invert” corrected on L432; nor “homogeneously mixed” corrected; Reference [41] still repeats Reference [34], etc.

Author Response

We thank the referees for their valuable comments. We have addressed all their comments carefully and revised the paper accordingly. The list of modifications and responses to referees’ comments are given below.

List of Main Modifications:

For clarity, all changes we have made are marked in the file entitled “Atmospheric_Ducts_and_Their_Electromagnetic_Propagation_Characteristics_in_the_Northwestern_South_China_Sea_diff.pdf”.

  1. Lower atmospheric duct models have been added in this paper.
  2. The reasons for using NCEP CFSv2 and ERA5 datasets have been added in this paper.
  3. The calculating equations from Result Section have been removed from the Result Section to the Method Section.
  4. The figure of the research flow chart of this paper has been removed from Method Section to the Introduction.
  5. Figure 12 has been added. (The electromagnetic propagation loss in different atmospheric ducts environments with radiation sources at frequencies of 6 GHz and 12 GHz:)
  6. The Figure 3/6/8/9/10/11/17/19/20 has been modified. The corresponding picture is Figure 2/7/8/9/10/11/18/20/21.
  7. The reference has been modified.
  8. The content of the abstract and summary has been reduced.
  9. Added comparison with existing research results.
  10. At the end of the introduction, the purpose and ideas of the research have been added.

Response to Reviewer 4 Comments

Dear reviewer,

We sincerely apologize for any inconvenience caused by the oversight in our previous communication. Regrettably, I contracted COVID-19 while working on the paper revision, and the revision time was relatively short, which resulted in a delay in addressing the comments provided. We deeply regret any inconvenience this may have caused.

Thank you for your understanding.

Point 1: Fig. 1 (c): in order not to confuse "trapping layer" and "duct thickness", right "trapping layer" instead of just "trapping''.

Response 1: This error has been corrected. (Page 6, Fig 3)

Point 2: lines 47-48 read: "The evaporation duct exists almost always over the ocean, but at different heights, generally occurring at a height of 40 m ..."; in fact, 40 meters are considered the max evaporation duct height, the world average evaporation duct height is much less than 40 m....

Response 2: This error has been corrected. (L44)  “… below 40 m in the atmosphere near the sea surface”

Point 3: lines 113-114 read: "the occurrence rate of the atmospheric ducts is the percentage of the duct frequency in the statistical time frequency", this is unclear.

Response 3: This error has been corrected. (L150-L151)

“The duct height ranges from 5 to 40 meters as a frequency of duct occurrence for the evaporation duct at a time.”

Point 4: Reference [34] does not refer to Bean and Datton; [34] repeats [41].

Response 4: This error has been corrected. (L138)

“…temperature, air pressure, and water vapor pressure be expressed as [41]:”

Point 5: Section 2.2: Rather than repeating known methods, it is better to cite one or two adequate references and say that the manuscript follows them.

Response 5: This error has been corrected. (Section 2.2)

Point 6: Provide Reference for formula (17).

Response 6: The reference for formula (17) has been added to this paper. (L152)

For the evaporation duct, the minimum trapping frequency of electromagnetic waves can be expressed by the vertical decrease rate of EDH and M as [42]:

Point 7: lines 317-318 read: "The electromagnetic wave radiation source must be located in the atmospheric ducts layer." This is not mandatory.

Response 7: This error has been corrected. (L322)

“Electromagnetic radiation sources situated within the atmospheric duct layer have a higher probability of being trapped.”

Point 8: Table 4 What is the beamwidth?

Response 8: 3dB beamwidth is 2°, which has been added in this paper. (table 4)

Point 9: Lines 327-330 read: "corrected refractive index" The quantities M and Mo are called Modified refractivity and Modified refractivity at height 0.

Response 9: This error has been corrected (L153)

“where M and M2 are called modified refractivity and modified refractivity at height 0 m”

Point 10: lines 430-432: "to invert the elevated duct" What do the authors mean under "invert"?

Response 10: This error has been corrected. “ to calculate the elevated duct” (L439)

Point 11: Line 457: Fig 18 instead of Figure 17; line 460: Fig.4 instead of Fig. 5

Response 11: This error has been corrected. (L470)

Point 12: Line 470: "... it is observed that as the elevation angle increases, the propagation loss decreases"; according to Fig. 19 (right) is the exact opposite...

Response 12: This error has been corrected. (L485)

“as the elevation angle increases, the propagation loss increases.”

Point 13: Sub-section 3.3 reads " Electromagnetic propagation in a homogeneously mixed duct environment"; in fact, this is a medium with more than one M inversion, not a "homogeneously mixed duct".

Response 13: This error has been corrected. “Electromagnetic propagation in hybrid duct environment”.

Reviewer 5 Report

Thanks for your replies to my comments that help me to better understand. I'm a bit confused about which of them are addressed in the final paper (the pdf you have attached with track changes is not so clear in this sense). But, finally, thanks for your effort in improving a bit this interesting paper. Good luck for your research in this area.

Author Response

Apologies for any inconvenience caused to your reading. Please see the attachment for the revision record I have added. Thank you.
